

# Speciated On-line PM$_1$ from South Asian Combustion Sources: *Part I, Fuel-based Emission Factors and Size Distributions*

**J. Douglas Goetz[1,a], Michael R. Giordano[1], Chelsea E. Stockwell[2,b], Ted J. Christian[2], Rashmi Maharjan[3,c], Sagar Adhikari[3,4], Prakash V. Bhave[4,d], Puppala S. Praveen[4], Arnico K. Panday[4], Thilina Jayarathne[5,] Elizabeth A. Stone[5,6], Robert J. Yokelson[2], Peter F. DeCarlo\*[1,7]**

[1] Drexel University, Department of Civil, Architectural, and Environmental Engineering, Philadelphia, PA, USA
[2] University of Montana, Department of Chemistry, Missoula, MT, USA
[3] MinErgy Pvt. Ltd, Lalitpur, Nepal
[4] International Centre for Integrated Mountain Development (ICIMOD), Lalitpur, Nepal
[5] University of Iowa, Department of Chemistry, Iowa City, IA, USA
[6] University of Iowa, Department of Chemical and Biochemical Engineering, Iowa City, IA, USA
[7] Drexel University, Department of Chemistry, Philadelphia, PA, USA
a now at: Laboratory for Atmospheric and Space Physics, University of Colorado at Boulder, Boulder, CO, USA
b now at: Chemical Sciences Division, NOAA Earth System Research Laboratory, Boulder, CO, USA
c now at: Clean Energy Nepal, Lalitpur, Nepal
d now at: Citizen's Climate Lobby, Raleigh, NC, USA

*Correspondence to:* Peter F. DeCarlo (pfd33@drexel.edu)



**Abstract.** Combustion of biomass, garbage, and fossil fuels in South Asia has led to poor air quality in the region and has uncertain climate forcing impacts. On-line measurements of submicron aerosol ($PM_1$) emissions were conducted as part of the Nepal Ambient Monitoring and Source Testing Experiment (NAMaSTE) to investigate and report emission factors (EFs) and vacuum aerodynamic diameter ($d_{va}$) size distributions from prevalent but poorly-characterized combustion sources. The on-line aerosol instrumentation included a "mini" aerosol mass spectrometer (mAMS) and a dual-spot 8-channel aethalometer (AE33). The mAMS measured non-refractory $PM_1$ mass, composition, and size. The AE33 measured black carbon (BC) mass and estimated light absorption at 370 nm due to organic aerosol, or brown carbon. Complementary gas-phase measurements of carbon dioxide ($CO_2$), carbon monoxide (CO), and methane ($CH_4$) were collected using a Picarro Inc. cavity ring down spectrometer (CRDS) to calculate fuel-based EFs using the carbon mass balance approach. The investigated emission sources include open garbage burning, diesel-powered irrigation pumps, idling motorcycles, traditional cookstoves fueled with dung and wood, agricultural residue fires, and coal-fired brick-making kilns, all of which were tested in the field. Open garbage burning emissions, which included mixed refuse and segregated plastics, were found to have some of the largest $PM_1$ EFs (3.77-19.8 g $kg^{-1}$) and the highest variability of the investigated emission sources. Non-refractory organic aerosol (OA) size distributions measured by the mAMS from garbage-burning emissions were observed to have lognormal mode $d_{va}$ values ranging from 145-380 nm. Particle-phase hydrogen chloride (HCl) was observed from the open garbage burning and was attributed to the burning of chlorinated plastics. Emissions from two diesel-powered irrigation pumps with different operational ages were tested during NAMaSTE. Organic aerosol and BC were the primary components of the emissions and the OA size distributions were centered at ~80 nm $d_{va}$. The older pump was observed to have significantly larger $EF_{OA}$ than the newer pump (5.18 g $kg^{-1}$ compared to 0.45 g $kg^{-1}$) and similar $EF_{BC}$. Emissions from two distinct types of coal-fired brick making kilns were investigated. The less-advanced, intermittently-fired clamp kiln was observed to have relatively large EFs of inorganic aerosol, including sulfate (0.48 g $kg^{-1}$) and ammonium (0.17 g $kg^{-1}$), compared to the other investigated emission sources. The clamp kiln was also observed to have the largest absorption Ångström exponent (AAE = 4) and organic carbon (OC) to BC ratio (OC:BC = 52). The continuously-fired zigzag kiln was observed to have the largest fraction of sulfate emissions with a $EF_{SO4}$ of 0.96 g $kg^{-1}$. Non-refractory aerosol size distributions for the brick kilns were centered at ~400 nm $d_{va}$. The biomass burning samples were all observed to have significant fractions of OA and non-refractory chloride and, based on the size distribution results, the chloride was mostly externally mixed from the OA. The dung-fueled traditional cookstoves were observed to emit ammonium suggesting that the chloride emissions were partially neutralized. In addition to reporting EFs and size distributions, aerosol optical properties and mass ratios of OC to BC were investigated to make comparisons with other NAMaSTE results (i.e. on-line photoacoustic extinctiometer (PAX) and off-line filter-based), and existing literature. This work provides critical field measurements of aerosol emissions from important yet under-characterized combustion sources common to South Asia and the developing world.

## 1 Introduction

South Asia is a culturally and geographically diverse region that is inhabited by nearly 25% of the world's population (United Nations, 2014). Although rapid urbanization is occurring throughout South Asia (Ellis and Roberts, 2016), much of the population lives in rural areas with limited access to public utilities (Palit and Chaurey, 2011;Bhattacharyya, 2007). Because of limited or inconsistent utility supplies, solid biofuels (e.g. wood, charcoal, agricultural residue, dung) are widely used in the region for residential cooking and heating and often in the indoor environment (Winijkul and Bond, 2016;Streets et al., 2003;Pandey et al., 2014;World Health Organization, 2006). Biofuels are also used throughout South Asia in the industrial sector





for brick making, in agricultural processing, and other activities (Pandey et al., 2014). Because of the atmospheric emissions from the combustion of solid fuels, heavy biofuel use in South Asia has air quality implications that range from indoor exposure (Chen et al., 1990) to regional outflow (Lawrence and Lelieveld, 2010), and leads to uncertain climate forcing impacts (Ramanathan et al., 2005;Venkataraman et al., 2005). In addition to biofuel, solid and liquid fossil fuel combustion from on-road vehicles, generators, diesel pumps, brick kilns, and coal-fired power generation are important trace gas and aerosol emission sources in the region (Lawrence and Lelieveld, 2010;Pandey et al., 2014;Reddy and Venkataraman, 2002).

The combustion of solid fuels (e.g. biomass, dung, coal) is often inefficient and has been observed to emit varying and often harmful levels of aerosols and trace gases. Fine aerosol emissions ($PM_{2.5}$) from solid fuel burning contain organic compounds, black carbon (BC), inorganic ions ($SO_4^{2-}$, $NO_3^-$, $Cl^-$, $NH_4^+$), and trace metals (Sheesley et al., 2003;Shahid et al., 2015;Roden et al., 2009;Kortelainen et al., 2015;Jayarathne et al., 2018;Bruns et al., 2015). Additionally, aerosol emissions from solid fuels contain polycyclic aromatic hydrocarbons (PAHs), which are known carcinogens (Sheesley et al., 2003;Jayarathne et al., 2018;Bruns et al., 2015;Chen et al., 2005). Gaseous emissions from solid fuel burning include carbon dioxide ($CO_2$), carbon monoxide (CO), methane ($CH_4$), non-methane organic compounds (NMOC), and other compounds such as nitrogen oxides ($NO_x$) and inorganic acids (Stockwell et al., 2014;Stockwell et al., 2015;Stockwell et al., 2016). In the indoor environment the use of solid fuels for cooking and heating leads to high levels of exposure to the above mentioned aerosols and trace gases (Chen et al., 1990) and poor indoor air quality from solid-fuel burning is one of the leading factors that contributes to the global burden of disease (Fullerton et al., 2008;Chafe et al., 2014;Agrawal and Yamamoto, 2015;World Health Organization, 2006). In South Asia, Lim et al. (2012) ranked household air pollution from solid fuel burning as the primary risk factor for populations in the region. The health impacts from poor indoor air quality due to solid fuel combustion demonstrate the importance of understanding emissions to quantify and potentially mitigate exposure.

On the local and regional scale, solid biofuel burning and fossil fuel combustion as well as other sources like open garbage burning and mineral dust have significantly degraded the air quality in South Asia. For example, many cities in India greatly exceed the U.S. National Ambient Air Quality Standards (NAAQS) for $PM_{10}$ (aerosol <10 μm) and nitrogen dioxide ($NO_2$) (Guttikunda and Goel, 2013;Guttikunda et al., 2014). In the Kathmandu Valley in Nepal, emissions of $PM_{10}$, volatile organic compounds (VOCs), and other pollutants from the above mentioned combustion sources combined with topography-induced entrapment have led to poor air quality (Panday and Prinn, 2009;Sarkar et al., 2016) and the formation of secondary pollution like ozone ($O_3$) (Putero et al., 2015). On a broader scale, the densely populated Indo-Gangetic Plain (IGP) region is a major source of atmospheric brown clouds that are formed from persistent anthropogenic aerosol emissions that are confined from mixing vertically due to wintertime boundary layer dynamics (Gautam et al., 2007;Nair et al., 2007). Outflow of wintertime atmospheric brown clouds has atmospheric impacts throughout South Asia and regions downwind (Gustafsson et al., 2009;Ramanathan et al., 2005). Lelieveld et al. (2001) found that winter monsoonal outflow from South Asia affects air quality over an area of 10 million $km^2$. Additionally, there is evidence that aerosol outflow from South Asia impacts regional climate through direct and indirect radiative forcing, which is thought to lead to stabilization of the troposphere, changing monsoonal patterns, and retreat of Himalayan glaciers (Lawrence and Lelieveld, 2010).

Regional emission inventories have shown that South Asia is responsible for a large portion of the aerosol emissions from the Asian continent and that regional emissions have been increasing. In a recent emission inventory, Li et al. (2017) estimated that South Asia was responsible for nearly 39% of Asian organic aerosol emissions and 35% of the BC emissions in the year 2010, and that South Asian emissions of both species increased by 13% and 25%, respectively, between 2006 and 2010. Although aerosol emissions from South Asia are known to be prevalent compared to other parts of Asia, the relative contributions of different source sectors to the regional aerosol loading remains uncertain. One of the early Asian emissions inventories, Streets et



al. (2003), found that of the regions in Asia, BC and organic aerosol emissions from South Asia had the highest percentage of uncertainty, which resulted from unknowns about biomass burning (e.g. biofuels and agricultural residue burning) emissions and liquid fuels consumption in the region. The unknowns in solid biofuel emissions in South Asia have produced significant differences between bottom-up and top-down estimates of BC emissions as well as differences between inventories that weigh

the relative contribution of biomass burning and fossil fuel combustion to BC emissions (Lawrence and Lelieveld, 2010). Uncertainty concerning biofuel emissions is largely due to the fact that emission sources which are prevalent in South Asia are not well characterized, both chemically and by quantity.

The above background reveals that aerosol emissions from biomass and fossil fuel combustion associated with prevailing sources in South Asia need to be further investigated. Better characterizing the emissions from South Asian combustion sources can aid

in understanding the impacts of residential exposure, provide key insights for local and regional air quality management, and constrain uncertainty about climate impacts. The goal of this study is to investigate aerosol emissions from prevalent sources found in South Asia and to provide some regional context for emission inventories. This study will focus on speciated submicron aerosol ($PM_1$) emission factors (EFs) and size distributions of primary aerosol emissions measured using on-line techniques during the Nepal Ambient Monitoring and Source Testing Experiment (NAMaSTE) that took place in Nepal in 2015. The results

complement other NAMaSTE works that made measurements simultaneously in the same plumes, but often without ideal spatial collocation or temporal overlap, at the tested emission sources. Stockwell et al. (2016) provides fuel-based EFs of $CO_2$, $CO$, $CH_4$, many other trace gases, BC, and aerosol absorption, and some additional aerosol optical properties. Jayarathne et al. (2018) collected filter-based $PM_{2.5}$ measurements and determined EFs for $PM_{2.5}$ mass, organic and elemental carbon, water-soluble organic carbon, inorganic ions, select metals, and organic molecular markers. In addition to providing stand-alone results, this

work aims to provide some comparisons of the aerosol EFs and optical properties that will guide the use of the complex NAMaSTE results as a whole.

## 2 Methods

The NAMaSTE campaign took place in April 2015 in and around the urbanized Kathmandu Valley, and in the rural Tarai region of southern Nepal, which is part of the IGP. As the name of the experiment suggests, NAMaSTE had two major components: (I)

ambient monitoring of aerosol and trace gases in the Kathmandu Valley and (II) characterization of aerosol and gas-phase emissions from combustion sources prevalent in South Asia. This work is part of the emissions testing portion of NAMaSTE with in-the-field, on-line $PM_1$ measurements of emission sources. A brief summary of the investigated emissions sources is given in Table 1, while detailed descriptions are provided by Stockwell et al. (2016). Additional source sampling was planned, but the campaign was cut short by the Nepal Gorkha earthquake on April 25, 2015. It should be noted that although the sample number

is limited and duplicate tests were not performed for many of the emission sources, this work provides critical real-world observations to the limited body of literature that is primarily comprised of laboratory measurements.

### 2.1 Experimental setup

Measurements were made by directly sampling the exhaust plume from each source with an attempt to sample at an adequate distance from the point of emissions (typically > 1 m) and away from the plume center to collect cooled and diluted emissions.

Emissions were sampled from well-mixed regions of the plumes instead of directly from the point of emissions in order to obtain more atmospherically relevant gas-particle partitioning of an emission plume, and for cookstoves, to simulate indoor ambient exposure. For example, in residences with cookstoves, sampling took place at the far end of the kitchen or by sampling from an



open eave in the building. Sample air was collected through ¼" copper tubing of varying length (1 – 5 m), which depended on site accessibility, that was connected to an on-line aerosol and gas sampling system. The longest inlet length was implemented at the forced-draft zigzag kiln to collect emissions from downwind of an 8.5 m tall chimney. The on-line sampling system was either setup in the bed of the truck that transported the equipment or was setup in a safe location near the emission source. The

system was powered using a gasoline generator, which was placed downwind of the emission source in each experiment typically at a distance of ~15 m. It should be noted that an effort was made to place the sampling inlet at each sampling site as close as possible to the sampling inlets of Jayarathne et al. (2018);Stockwell et al. (2016). However, due to the low mobility of the on-line sampling platform used in this work compared to the other sampling platforms the inlets were not always collocated, but there was typically no more than a meter of separation. The separation of the aerosol inlets could have led to some

differences in ambient dilution of the emission plumes prior to entering the sampling inlets. In addition, the time period during which the various approaches were deployed was not always exactly the same. This increased the sampling coverage, but also the uncertainty in some comparisons.

The on-line sampling system was made up of two major components the undiluted flow system and the diluted flow system. A diagram of the sampling system can be found in the supporting information (Fig. S1). The undiluted flow system included

aerosol-free $CO_2$ monitoring. The diluted flow system was comprised of an inline HEPA filter bypass (for periodic zero calibrations), a Dekati Ltd. Axial Diluter (DAD-100), and a $PM_{2.5}$ cyclone, which fed to on-line aerosol and gas-phase instrumentation. Excess flow was controlled with a needle valve and diaphragm pump. The axial diluter was calibrated to provide 15.87 SLPM of dilution air at a pressure of 3500 mbar. All dilution air was obtained from ambient background air outside the plume at each site and filtered to remove aerosols prior to injection in the dilution system. Dilution factors were

calculated in the field by monitoring sample and dilution volumetric flow rates and later were verified using molar ratios of $CO_2$ from the undiluted flow systems to $CO_2$ from the diluted flow system. The axial diluter was typically operated at a dilution factor between a range of 1:1 to 22:1, with an average of ~10:1. The large range of dilution factors used in this experiment was due to the varying downwind distance and source strength between the investigated emission sources. Lab experiments conducted after NAMaSTE found that the sampling system has a $PM_1$ transmission rate of 97.6% for ammonium nitrate aerosol. The measured

system transmission rate was for dilution factors from 1:1 to 15:1. Transmission was determined to be independent of the dilution factor for non-volatile aerosol.

### 2.2 Instrumentation

### 2.2.1 mAMS

Using a mini Aerosol Mass Spectrometer (mAMS; Aerodyne Research Inc.), the mass, composition, and size of sub-micron

aerosol that volatilize under vacuum at 600 C (operationally defined as non-refractory species) was measured. The measured non-refractory aerosol components include organics, sulfates, nitrates, chlorides, and ammonium. The mAMS is a version of the Aerodyne Research, Inc. Aerosol Mass Spectrometer that is functionally similar to the compact Time-of-Flight-AMS (c-TOF-AMS) (Drewnick et al., 2005), but with a smaller time-of-flight spectrometer and a smaller vacuum chamber with a pump system that utilizes a single split flow turbo molecular pump. The mAMS has the same body, turbo pump system, and v-mode time-of-

flight mass spectrometer as the Time-Of-Flight Aerosol Chemical Speciation Monitor (TOF-ACSM) (Fröhlich et al., 2013), but contains a chopper system (Jayne et al., 2000) and a more advanced data acquisition card for particle time-of-flight sizing. The mAMS used in this work operates with a pseudo-random multi-slit chopper system (ePTOF) that has increased signal to noise (~50% particle throughput) compared to single slit chopper systems with ~2% throughput, and employs Hadamard Transform for



signal inversion (Campuzano Jost, 2014). Because of its enhanced throughput, the ePTOF is ideal for sampling highly-transient concentrations like those encountered during emissions source testing. ePTOF data reports mass concentration of chemical species as a function of vacuum aerodynamic diameter ($d_{va}$), similar to the PTOF data in other AMS instruments (DeCarlo et al., 2004;Jimenez et al., 2003).

The mAMS operated in both mass spectrum mode (MS) and particle time-of-flight mode (ePTOF) for the entirety of the source experiments with the exception of several Nepal Renewable Energy Test Station (RETS) laboratory cooking fires in which the chopper system was not operational. The MS and ePTOF sampling alternated every 5 seconds and 5 second integrated data from both modes was saved every 10 seconds for an effective sampling rate of 0.10 Hz. Mass spectra were acquired from 10 - 300 m/z for all data collected. For the entirety of the measurements, the mAMS vaporizer was operated at 600°C. It should be noted that

although we categorize the aerosol detected by the mAMS as submicron, transmission of aerosol between 1 μm to 2.5 μm through the aerodynamic lens of the instrument does occur and similar lenses have been characterized to have 2.5 um transmission efficiencies of less than 50% (Zhang et al., 2004).

Ion efficiency calibrations were conducted twice in April while the instrument was in Nepal. Other in-country calibrations of the mAMS were canceled because of the Gorkha Earthquake. Velocity calibrations were conducted using polystyrene latex spheres

(PSLs) at sizes between 50 – 800 nm after the campaign at the Drexel lab at inlet pressures of 0.76 bar and 1.01 bar. The velocity calibrations were conducted at the above pressures to simulate particle time-of-flight velocities at the atmospheric pressures observed in the high-altitude Kathmandu Valley (0.76 bar) and in the Tarai Plains (1.01 bar).

All data processing and analysis was done in Igor Pro 6.3 (Wavemetrics, Lake Oswego, OR) using standard TOF-AMS analysis software SQUIRREL v1.57I and PIKA v1.16I. The initial mass spectral separation into aerosol components was performed using

the standard fragmentation table (Allan et al., 2004), and PAH signal was identified using the method of Dzepina et al. (2007). Although the mAMS is not a high resolution mass spectrometer, the resolution is sufficient for the separation of some key ions at the same nominal mass to charge (DeCarlo et al., 2006). The mass spectral data were processed using high-resolution peak fitting in the PIKA module to reduce fragmentation table errors due to high organic loading. High-resolution treatment of raw mass spectral data from the compact Time-of-Flight MS has previously been performed by other researchers using the TOF-ACSM

with an estimated resolving power (M/ΔM) of ~600 (Fröhlich et al., 2013). A collection efficiency of 0.5 was applied to all of the data sets (Matthew et al., 2008). Source-test-specific detection limits of the aerosol species measured by the mAMS were calculated using data from the HEPA bypass filter periods, which occurred at least twice per emissions test for a period of 10 minutes each. Detection limits are defined as 3σ of the combined filter periods for each source experiment.

**2.2.2 Aethalometer**

A Magee Scientific AE33 aethalometer was used to measure light-absorbing, carbonaceous PM concentrations, absorption coefficients, and absorption Ångström exponents. The AE33 is a dual-spot, filter-based monitor that measures light attenuation by particles on a Teflon filter tape at 8 wavelengths (370, 470, 525, 590, 660, 880 and 950 nm) and, unlike previous aethalometer models, the AE33 allows for real-time filter-loading compensation (Drinovec et al., 2015). Light-scattering artifacts, which can be misinterpreted as light absorption by the filter-attenuation-detection methods of the AE33, were corrected using a scattering

coefficient (C) developed by Schmid et al. (2006) and more recently implemented for the intercomparison of commercial optical instrumentation by Segura et al. (2014). The scattering correction is shown as a function of wavelength (λ) in Eq. 1, where C is the total scattering coefficient for the given wavelength, C* is the multiple scattering coefficient for the given filter material, $m_s$ is the fraction of aerosol scattering erroneously interpreted as absorption for purely scattering aerosol, and $\omega_o$ is the single scattering albedo (SSA) of the sampled aerosol.



$$C(\lambda) = C^*(\lambda) + m_s(\lambda)\frac{\omega_o}{1-\omega_o} \qquad (1)$$

The $C^*$ and $m_s$ for each wavelength are taken from work by Arnott et al. (2005). The SSA was estimated from average photoacoustic extinctiometer (PAX) measurements of scattering at 405 nm and 870 nm for each emission source by Stockwell et al. (2016). Because $\omega_o$ was available at only two wavelengths, C is only implemented at the nearest wavelength channels measured by the AE33 (370 and 880 nm) to calculate aerosol absorption for each emissions test. Aerosol absorption at each wavelength was calculated using equation 17 of Drinovec et al. (2015), which is the same equation used internally by the AE33 to calculate the dual-spot-corrected mass output.

The absorption Ångström exponent (AAE) was calculated for each emission source using Eq. 2 based on the test-integrated and scattering-corrected absorption coefficients at 370 and 880 nm. Source-specific AAE results can be found in Section 3.6.

$$AAE = -\frac{\log(B_{abs,370}/B_{abs,880})}{\log(370nm/880nm)} \qquad (2)$$

The AAE, which is a measurement of the wavelength dependence of light absorption by aerosols, has been observed to be ~1 for externally-mixed pure BC (i.e., soot) emissions and >1 when there is enhanced absorption at short wavelengths associated with BrC emissions from biomass burning sources or from internal mixing with non-absorbing material (Lack and Langridge, 2013;Olson et al., 2015;Stockwell et al., 2016;Wu et al., 2016).

Absorption at 880 nm ($B_{abs,880}$) was used to quantify black carbon (BC) or soot aerosol. Following the methodology of Stockwell et al. (2016), the absorption at 370 nm not attributed to BC was used to quantify light–absorbing organic carbon aerosol, or brown carbon (BrC). Assuming an AAE of 1 for externally-mixed BC, where light absorption is proportional to frequency, absorption due to BrC at 370 nm ($b_{abs,BrC}$) is therefore computed as $b_{abs,370}$ - $2.37\ b_{abs,880}$. Excess absorption at 370 nm by multi-channel aethalometers has previously been attributed to BrC emissions from biomass burning (Olson et al., 2015;Wang et al., 2012) and uncertainties in the attribution are discussed elsewhere (Pokhrel et al., 2017). Mass concentrations were calculated using mass-absorption cross sections of 7.77 $m^2\ g^{-1}$ and 18.47 $m^2\ g^{-1}$ for 880 and 370 nm, respectively, based on recommendations by Drinovec et al. (2015). HEPA-bypass periods were used to zero-calibrate the BC mass. The AE33 operated at a sampling rate of ~1 Hz and was averaged to 0.10 Hz to match the sampling scheme of the mAMS. All BC detected by the AE33 in this work was assumed to be submicron particles based on the morphology of fresh biomass-burning and fossil-fuel emissions observed in other studies (China et al., 2013;Gong et al., 2016;Torvela et al., 2014).

### 2.2.3 Gas-phase instrumentation

Gas-phase instrumentation included a Picarro Cavity Ring Down Spectrometer (CRDS) Model G2401, a Licor $CO_2$ and $H_2O$ monitor (Li840A), a Vaisala $CO_2$ monitor (GMP343), and a Gaslab Inc. high range $CO_2$ monitor (Fig. S1). The Picarro CRDS was used as our primary measure of diluted $CO_2$, CO, and $CH_4$ while the Vaisala was used as a backup measure of diluted $CO_2$. The Licor $CO_2$ monitor was the primary undiluted $CO_2$ measurement. It also served as a calibration reference for the other $CO_2$ monitors because calibrations with a laboratory standard could not be conducted while the instruments were in Nepal and the Licor monitor had been factory-calibrated only 2 months before the campaign and only operated with aerosol free ambient air. The $CH_4$ and CO measurements could not be calibrated because of contaminants in the calibration gases available in Nepal. However, comparisons with the canister measurements taken at similar times by Stockwell et al. (2016) and during a collocated ambient sampling period reveal that $CH_4$ concentrations from the Picarro were 1% lower than the whole air sample values and that Picarro CO concentrations were up to 30% higher.

### 2.3 Combustion metrics





For each source experiment, time-resolved and test-integrated emission factors (EF) were calculated in units of g per kg of fuel using the carbon mass balance approach (Ward, 1990). In Eq. (3), $EF_n$ is the EF of aerosol species $n$ (g kg$^{-1}$), $f$ is the fraction of fuel mass consisting of carbon, and $\Delta n$ and $\Delta[C]_i$ are the excess concentrations of aerosol species $n$ and gas-phase species $i$, respectively, above the ambient background.

$$EF_n = f \frac{\Delta n}{\Delta[C]_{CO_2} + \Delta[C]_{CO} + \Delta[C]_{CH_4}} \quad (3)$$

In the denominator of Eq. (3), we implicitly assume that all carbon emitted in the forms of NMOC and PM during our source tests are minor when compared to the total carbon emitted in the forms of $CO_2$, CO, and $CH_4$ (Akagi et al., 2011;Stockwell et al., 2016;Stockwell et al., 2015). For example, $PM_1$ carbon mass was estimated to be 0.58±0.43% of the combined carbon mass of $CO_2$, CO, and $CH_4$, emitted by the investigated sources.. Missing carbon should bias EF upwards by less than 1-2% in general.

The $f$ for each fuel (see Tables S1-S4 of the supporting information) was obtained from fuels collected during NAMaSTE where possible or the most appropriate literature values otherwise (Stockwell et al., 2016).

Another metric used in this work is the modified combustion efficiency (MCE), defined here as the ratio $\Delta CO_2/(\Delta CO_2 + \Delta CO)$. This is a useful metric provided that >90% of emissions are comprised of CO and $CO_2$. Because the Picarro CRDS measurements of CO consistently exceeded the canister measurements and could not be calibrated, MCE values shown in this

work are from the Fourier transform infrared spectroscopy (FTIR) test-integrated measurements made by Stockwell et al. (2016). Time-resolved MCE derived from the Picarro CRDS are therefore only used to fill data gaps or as secondary evidence of trends.

Organic carbon (OC) is used to compare our results with studies that measured $PM_{2.5}$ $EF_{OC}$ from off-line, filter-based, thermal-optical methods. Because the OA mass measured with the mAMS includes OC as well as other non-carbon organic mass (e.g. hydrogen, oxygen, nitrogen, and sulfur atoms), OC in this work is estimated based on oxygenation of the bulk, non-refractory

OA. Assuming oxygen comprises the majority of non-carbon organic mass, the mass fraction of total OA that is detected at m/z 44 ($f_{44}$) has been found to be a useful proxy for the oxygenated organic mass of aerosol when high-resolution AMS data are not available (Aiken et al., 2008). Oxygen-to-carbon atomic ratios (O:C) were estimated based on work by Canagaratna et al. (2015) using test-integrated $f_{44}$ values from each emission test. Organic aerosol to organic carbon ratios (OA:OC), which are sometimes referred to elsewhere in the literature (Simon et al., 2011) as organic mass-to-carbon ratios (OM:OC), were determined based on

the linear relationship between OA:OC and O:C found by Aiken et al. (2008). The $f_{44}$, O:C, and OA:OC values for each emission source can be found in Table 2. OA:OC was estimated to have an average uncertainty of ±5.3% for the investigated emission sources based on error in $f_{44}$ (Table 2). It should be noted that the large $f_{44}$ associated with the coal-fired zigzag brick kiln was due to the presence of nitrogen-containing organic ions at m/z 44 ($C_2H_6N^+$) and not due to oxygenated organics as discussed in the companion paper Goetz et al. (2018). Because the O:C of the zigzag kiln emissions could not be determined based on $f_{44}$, the

O:C ratio from the clamp kiln is used to approximate OC emission factors for the zigzag kiln emissions.

### 3 Results and discussion

This work combines the non-refractory PM1 measurements from the mAMS with the black carbon measurements from an Aethalometer to determine online PM1 fuel-based EFs and size resolved EFs from the field-tested emission sources provided in Table 1. Mass spectral profiles of aerosol emissions from the investigated combustion sources can be found in Part II of this

study (Goetz et al., 2018). A summary of the test-integrated, online, fuel-based $PM_1$ emission factors ($EF_{PM1}$) of the field-tested emission sources is given in Fig. 1. Tabulated EFs including percentiles, average and standard deviation, and test-integrated EFs for the field-tested emission sources shown in Table 1 can be found in the supporting information Tables S1-S4. The tables also contain MCE values for each field-tested emission source from Stockwell et al. (2016) and uncalibrated Picarro CRDS derived



MCE values from this work. Emission factors from the cookstoves tested at the RETS laboratory are not included in this work because of poor venting in the RETS lab that elevated background concentrations and elevated gas-phase concentrations to the dilution system, which produced unreliable results. The sum of all particulate components measured by the mAMS and aethalometer (non-refractory primary OA, sulfate, nitrate, chloride, and ammonium; and BC) are reported here collectively as PM$_1$. Polycyclic aromatic hydrocarbons are not included in the sum because they are already included as part of the OA total. Emissions of other refractory aerosol species that were not measured with the mAMS, (e.g. refractory OA, trace metals, mineral dust) and slow-vaporizing aerosol species are not included in the PM$_1$ calculation.

The largest EF$_{PM1}$ observed in this study were from open garbage burning and the diesel powered groundwater pumps. Plastic burning associated with open garbage burning had a EF$_{PM1}$ of 19.8 g kg$^{-1}$ of fuelEF$_{PM1}$ ranging from about 2.7 to 7.2 g kg$^{-1}$ were observed from open burning of other refuse (mixed and chip bags) and emissions from the diesel groundwater pumps. Biomass burning emissions from the field-tested cookstoves and agricultural residue burning generally had EF$_{PM1}$ values between 2.3 to 4.5 g kg$^{-1}$ with the exception of dung burning in the 1-pot traditional mudstove (1.8 g kg$^{-1}$) (Fig. 1). Generally, the PM$_1$ emissions from the above-mentioned sources were primarily comprised of OA followed by BC. The coal-fired brick kilns were observed to have lower EF$_{PM1}$ compared to the biofuel burning, and contain lower fractions of OA and BC and significantly larger fractions of sulfate.

Mass size distributions of the aerosol species observed by the mAMS were calculated at a range of 30 nm to 2 μm $d_{va}$ and normalized by the test integrated EF of each source to produce size-resolved EFs in mg per kg of fuel (Fig. 2 and Fig. 3 right axis). The size-resolved emission factors were also binned into aerodynamic diameter size cuts common with aerosol impact samplers (32, 56, 100, 180, 320, 560, 1000 nm) and plotted as stacked bars to produce Lundgren style cumulative size distributions (Kleeman et al., 1999) and normalized by the total non-refractory PM$_1$ emissions of each source (Fig. 2 and Fig. 3 left axis). The Lundgren style plots provide a better understanding of aerosol composition at each size and offers more interpretable results for model inputs compared to the species segregated continuous distributions. The size distributions of PM$_1$ emissions are important inputs for indoor exposure and lung deposition models, as well as important parameters for chemical transport models. The size-resolved results for open garbage burning, agricultural residue burning, and the field measured traditional mudstoves can be found in Fig. 2. Size-resolved emission factor of sources primarily associated with fossil fuel combustion can be found in Fig. 3. The following subsections contain a source-type examination of the EF results.

### 3.1 Open garbage burning

The three types of open garbage burning that were tested in NAMaSTE include mixed refuse, plastic, and metalized plastic or "chip bags" (Table 1). We sampled mixed refuse emissions in two separate burns and both mixes were comprised of unknown fractions of plastic bags, metalized plastic, food waste, paper, and yard waste that were collected from local sources. Mix 1, which was sourced and burned in the Kathmandu Valley, was slightly damp producing inefficient burn conditions with an average MCE of 0.937 (Stockwell et al., 2016). Mix 2 was residential waste burning sampled in the Tarai Plains at dry conditions and was more efficient than Mix 1 with an average MCE of 0.980 (Stockwell et al., 2016). The two mixes had a combined average EF$_{PM1}$ of 3.99 g kg$^{-1}$ with an approximate OA fraction of 0.50 and BC fraction of 0.48 (Fig. 1). Trace aerosol species were found to have EFs in mg kg$^{-1}$ of 66, 3, and 2 for chloride, PAHs, and nitrate, respectively. The two mixes differed in both OA and BC emissions. Mix 1 had an EF$_{OA}$ of 3.5 g kg$^{-1}$ and a low fraction of BC emissions with a EF$_{BC}$ of 0.19. Alternatively, Mix 2 emissions had a lower OA fraction with an EF$_{OA}$ of 1.35 g kg$^{-1}$ and an EF$_{BC}$ of 2.67 g kg$^{-1}$. The mixes were also found to have large variability in real-time emissions with combined EF$_{OA}$ having a 25$^{th}$ percentile of 0.45 g kg$^{-1}$ and a 75$^{th}$ percentile of 3.42 g kg$^{-1}$ (Fig. 1). Similar variability was also observed with EF$_{BC}$, with combined interquartile values ranging





from 0.20 to 3.55 g kg$^{-1}$ (Fig. 1). Additionally, the two mixes had distinct OA size distributions with Mix 1 having a lognormal-mode vacuum aerodynamic diameter, hereafter named "mode $d_{va}$", at 260 nm and Mix 2 having a mode $d_{va}$ at 145 nm (Fig. 2). The variability in emissions observed between the mixes and within individual burns demonstrates that open garbage burning is a difficult to characterize emission source because of the inherent heterogeneity of residential garbage and because of the

uncontrolled nature of open burning.

Open garbage burning is a globally important source of aerosol pollution, but there have been limited field measurements of open garbage burning EFs (Wiedinmyer et al., 2014). In NAMaSTE the filter-based measurements of Jayarathne et al. (2018) found an organic carbon (OC) EF for PM$_{2.5}$ of 8.42 ±0.63 g kg$^{-1}$ for Mix 2. Based on the estimated OA:OC for open garbage burning (1.361; Table 2) the OC EF (EF$_{OC}$) measured by Jayarathne et al. (2018) for Mix 2 was about 6 times greater the

combined average EF$_{OC}$ measured in this work of 1.46 g kg$^{-1}$. Filter-based measurements of open burning of landfills in Mexico, which are used as the primary EF$_{OC}$ resource by emissions inventories, reported MCE-dependent EF$_{OC}$ ranging from 2.13 g kg$^{-1}$ to 10.9 g kg$^{-1}$ with an average of 5.27 g kg$^{-1}$ (Christian et al., 2010). Although the EF$_{OC}$ range reported by Christian et al. (2010) is greater than our combined average, the mAMS Mix 1 EF$_{OC}$ was 2.57 g kg$^{-1}$ and within the lower limit of the Mexico garbage burning values and the upper percentiles were well within the upper limit of the literature values. The overlap in variability

between the off-line and on-line NAMaSTE EF$_{OC}$ results and Christian et al. (2010) demonstrate that, although there is likely heterogeneity in MCE and materials contained in the sampled open garbage burning piles, it is possible that a limited range of EFs exists for garbage burning that can be applied to regional emission inventories. However, differences in detection methods (i.e PM$_1$ vs PM$_{2.5}$), sampling procedures, or degrees of phase partitioning of semi-volatile organics (Lipsky and Robinson, 2006) cannot be removed as possible factors responsible (or partially responsible) for the differences observed between the on-line and

off-line results. This concept is specifically acute in regard to the NAMaSTE results where differences in sampling inlet locations and timing as well as differences in sample dilution could have played a part in the observed EF$_{OC}$ discrepancy. Further measurements of open garbage burning EF$_{OC}$ by on-line and off-line methods are needed to further explore the role of detection methods on OC results. Additionally, further measurements of aerosol emissions from open garbage burning with an emphasis on quantifying the contents of the garbage are needed to constrain compositional dependency of EF$_{OC}$.

In regard to BC emissions from open garbage burning, the on-line PAX derived EF$_{BC}$ from Stockwell et al. (2016) had similar trends between the two mixes as the EF$_{BC}$ measured in this work, with a larger EF$_{BC}$ from the higher MCE mix (6.04 g kg$^{-1}$) and lower EF$_{BC}$ from the lower MCE and damp mix (0.561 g kg$^{-1}$). The PAX damp mix value compares well with observations by Christian et al. (2010) who found an average elemental carbon EF of 0.646 g kg$^{-1}$ with burn conditions that produced an average MCE of 0.950. Based on the PAX and Mexico averages, Stockwell et al. (2016) suggest an upward revision of the literature

average EF$_{BC}$ might be appropriate and we also observed higher EF BC at least on mix 2. However, given the evidence that some relationship between the moisture content of the garbage and EF$_{BC}$ may exist, any investigation of compositional dependence on aerosol EFs from open garbage burning should be coupled with an investigation of moisture content. Finally, EF$_{BC}$ measurements should not be affected by phase partitioning and thus the factor of ten difference for the two mixed garbage burns observed by the PAX suggests that natural variability may be as, or more, important than differences in phase partitioning in

understanding the EF$_{OC}$ differences. The high variability suggests that more sampling by any technique is needed to increase coverage and that co-collection of fuel details that may help rationalize the variability should be emphasized.

As seen in Fig. 1, the segregated plastic burn was observed to have the largest EF$_{OA}$ (16.59 g kg$^{-1}$) of any source investigated in the study and contained some of the largest EFs for chloride (0.502 g kg$^{-1}$) and PAHs (23 mg kg$^{-1}$). Plastic burning had a EF$_{BC}$ of 2.73 g kg$^{-1}$ and a sulfate EF (EF$_{SO4}$) of 0.015 g kg$^{-1}$ (Fig. 1). The OA emissions were observed to have a size distribution with a

mode $d_{va}$ of 280 nm (Fig. 2).



The metalized plastic, or foil "chip bags", burning was observed to have a median $EF_{PM1}$ of 5.8 g kg$^{-1}$ and was comprised of 60% OA and 40% BC, with nominal quantities of inorganic aerosol and PAH (Fig. 1). Unlike the other open garbage burning tests there were relatively low emissions of particulate chloride and Stockwell et al. (2016) did not observe gas-phase hydrogen chloride (HCl) above detection limits. The size distribution of the metalized plastic emissions had the largest mode of the

sampled open garbage burning with a $d_{va}$ of 380 nm and large fraction of aerosol with diameters greater than 560 nm (~28%; Fig. 2).

Chloride in the form of gaseous HCl and water soluble particulate Cl- has been observed from open garbage burning in Mexico and has been attributed to the combustion of polyvinyl chloride (PVC) plastic (Christian et al., 2010). The high levels of chloride observed in the plastic burning emissions and mixed refuse emissions that were not observed with metalized plastic burning are

therefore likely from the combustion of PVC plastic. Analysis of average mass spectra from open garbage burning indicates that the non-refractory chloride measured by the mAMS was between 80-85% particle phase HCl. A comparison with undiluted gas-phase EFs from Stockwell et al. (2016) shows that the particle phase HCl EF was 1.2%, 2.5%, and 0.4% of gas-phase HCl emission factor for Mix 1, Mix 2, and plastic burning, respectively. The small quantity of particle phase chloride under controlled dilution conditions compared to gas-phase HCl under less dilute conditions suggests that condensation of HCl to the

particle phase is small in fresh emissions from open garbage burning, but evidence suggests HCl gas migrates overwhelmingly to the particles on slightly longer time scales (Liu et al., 2016;Stockwell et al., 2014).

**3.2 Engine exhaust**

Sampling of engine exhaust from idling motorcycles and diesel powered, groundwater crop irrigation pumps took place during NAMaSTE (Table 1). Testing of gas and diesel generators also took place during the campaign (Jayarathne et al.,

2018;Stockwell et al., 2016), but the generators were not sampled by the on-line sampling system described in this work. Two ~5 kVA irrigation pumps were sampled including a Kirloskar (model unknown) that had been in operation for 3 years and a Field Marshall model R170a that had been purchased within three months of the emissions test. The older pump (Pump 1) was observed to have a $EF_{PM1}$ of 7.24 g kg$^{-1}$ with an OA fraction of 0.71 and a BC fraction of 0.29 (Fig. 1). Inorganic aerosol was not observed from Pump 1. The median organic mass distribution of Pump 1 had a mode diameter of ~80 nm $d_{va}$ and nearly 38% of

the organic emissions were found in the 56 to 100 nm size bin (Fig. 3). The newer pump, Pump 2, was observed to have a lower $EF_{PM1}$ than Pump 1 with a value of 2.71 g kg$^{-1}$. Pump 2 emissions had an organic fraction of 0.16, a BC fraction of 0.83, and a nominal fraction of sulfate (Fig. 1). The Pump 2 organic size distribution was similar to Pump 1 (mode diameter = 75 nm $d_{va}$), but with a larger fraction of aerosol found in the 56 – 100 nm bin (Fig. 3). Emissions of PAHs were observed from Pump 2 with an EF of 6 mg kg$^{-1}$ and were not observed from Pump 1. The PAH EF from Pump 2 was approximately equivalent to EFs that

have previously been observed from heavy-duty diesel trucks in the United States (Marr et al., 1999). Stockwell et al. (2016) found that Pump 1 and Pump 2 had an MCE of 0.987 and 0.996, respectively. Because the diesel fuel used by both irrigation pumps was likely sourced from the same depot of the Nepal Oil Corporation and, therefore was similar in composition, the large differences observed between the investigated pumps were likely due to differences in efficiency induced by operational age or by model.

The filter-based measurements of Jayarathne et al. (2018) observed larger organic emission factors from both pumps with an average $EF_{OC}$ of 5.45 g kg$^{-1}$, compared to an average of 2.12 g kg$^{-1}$ from the on-line mAMS measurements. The filter-based results, however, measured significantly lower EC compared to the on-line $EF_{BC}$ in this work and the 405 nm photoacoustic measurements of Stockwell et al. (2016). The lower EC may have been because the engine start-up emissions were not sampled with the filters. Although Stockwell et al. (2016) did not observe the same increase in BC with Pump 2 and in fact observed a



13% decrease in BC between Pump 1 and Pump 2. The FTIR measurements, however, did observe a decrease in gas-phase organic EFs from Pump 1 to Pump 2 and there was a 200% increase in nitric oxide (NO) between the two pumps (Stockwell et al., 2016). The decreased $EF_{OA}$ combined with elevated MCE and elevated emission factors of NO, BC, and PAHs associated with the newer irrigation pump suggests that the air quality and climate impacts of irrigation pumps can likely change over the lifetime of a pump. The differences between the three NAMaSTE detection and sampling methods for black carbon were not tested carefully in this work, but can produce differences in measured mass (Hitzenberger et al., 2006;Watson and Chow, 2002). To our knowledge NAMaSTE is the first to characterize emissions from diesel powered irrigation pumps during in field use. Although there is uncertainty in the extent of BC and OA emissions, averaging all the NAMaSTE sampling implies that large $EF_{PM1}$ (~8.17 ± 2.55 g kg$^{-1}$) are associated with the tested pumps and that ground water pumping for irrigation could be an important source of aerosol pollution in rural South Asia. This is especially true in countries like India where the number of diesel powered pumps has increased over recent decades (Mukherji, 2008)

Four idling, gasoline-powered, 4-stroke motorcycles were sampled in this study. Sampling included 2 Honda CBZs, a Bajaj Pulsar, and a Bajaj Discover. The vehicles were sampled directly after servicing. Pre- and post-service sampling of idling motorcycles was part of the NAMaSTE sampling plan, but the mAMS was not operational for the pre-service period of the experiment. Information about the reduction of aerosol and gas-phase emissions resulting from servicing can be found in the NAMaSTE companion papers (Jayarathne et al., 2018;Stockwell et al., 2016). The combined post-servicing results from the four motorcycles investigated in this work indicate that organics were the only aerosol component observed above the background. Although not shown in Fig. 1, the observed $EF_{OA}$ were two orders of magnitude lower than OC observations by (Jayarathne et al., 2018) and observations by U.S.-based motorcycle studies (Bond et al., 2004). The low OA emission factors observed by this study are thought to be due to large and variable background CO, $CH_4$, and $CO_2$ at the motorcycle shop where testing was performed. The large unstable backgrounds were due to poor venting of emissions from the tested motorcycles and vehicle emissions from the adjacent, congested Kathmandu road that likely affected the gas-phase concentrations of the aerosol-free air injected into the dilution system. The $EF_{OA}$ for idling motorcycles derived from this work are therefore not given because of the unreliable gas-phase results needed for the carbon mass balance. Since we could remove particles, but not gases from the dilution air, the size distributions and mass spectra derived from the mAMS data could be corrected for unstable backgrounds. The mass size distributions indicate that the mode $d_{va}$ the motorcycle emissions was 107 nm and nearly 53% of the mass had a diameter between 56 nm and 180 nm (Fig. 3). Another study that investigated Asian motorcycle emissions found a similar size distribution range for $PM_{2.5}$, but observed a bimodal distribution with modes above and below 100 nm (Yang et al., 2005). A study of 4-stroke Asian motorcycles found that the $PM_{2.5}$ size distribution of idling motorcycles had the largest mode diameter of the operation cycles investigated and that the distributions shift to smaller sizes when the motorcycles were operated at 15 and 30 km per hour (Chien and Huang, 2010). Results from the above studies suggest that the OA size distributions we observed were at the middle to upper range of OA sizes for motorcycle emissions and that the mode of the distribution would likely shift to ultrafine sizes when the motorcycles are operated above idle.

### 3.3 Brick kilns

Brick kilns are a poorly characterized but important source of aerosol emissions in South Asia (Weyant et al., 2014). In NAMaSTE we investigated emissions from two kilns representing distinct classifications of brick kilns. In the Kathmandu Valley we sampled emissions from a batch-style clamp kiln which are typical of cottage industries across South Asia. In central Nepal we investigated emissions from a zigzag kiln which represent a modern kiln type found in the region. Although we were





not able to obtain a large sampling set, the results presented here and in other NAMaSTE works are some of the first to characterize the aerosol and gas-phase composition of brick kiln emissions in South Asia.

Clamp kilns are a traditional and inefficient brick-firing technology that use intermittent firing (single batch per firing) and are not designed with a chimney or draft system (Manadhar and Dangol, 2013). Because of the design of the clamp kiln, we sampled

fugitive emissions escaping from cracks at the top of the kiln. The kiln was co-fired with coal and hardwood. Over the ~4 hour time span in which mAMS and AE33 sampling took place the $EF_{PM1}$ was 1.759 g kg$^{-1}$ (Fig. 1). The $EF_{PM1}$ was comprised of 57% OA, 28% sulfate, 9.6% ammonium, 5.3% chloride, and <1% BC (Fig. 1). PAHs were not observed above detection limits. The speciated size distributions of the clamp kiln emissions show that the organic component had the largest mode diameter at 453 nm $d_{va}$ and the mode diameter of each inorganic component decreased in the same rank order as the fraction of PM$_1$ mass (Fig.

3). Chloride, for example, was found to have the lowest mode $d_{va}$, at 315 nm or ~140 nm smaller than the OA mode. Regardless of the differences in the estimated mode diameters between the aerosol species, nearly 87% of the non-refractory mass was found in a size range between 180 nm and 1000 nm. Additionally, aerosol mass signal from particles smaller than 100 nm $d_{va}$ was virtually zero for the clamp kiln (Fig. 3).

Zigzag brick kilns are a subset of fixed-chimney bull's trench kilns (FCBTK) and an established kiln type in Nepal (Manadhar

and Dangol, 2013). The kilns utilize continuous firing with bricks stacked in a zigzag pattern to optimize heat transfer efficiency and they use a forced or natural draft with a fixed chimney. In NAMaSTE we sampled from a forced-draft zigzag kiln. The kiln was fired with coal and sugar cane post-pressing residue (bagasse) was used as a starter fuel. The kiln was stoked periodically with the addition of coal through openings above the heated section of the kiln. More details on the operation of the investigated brick kilns and why they were chosen for sampling can be found in Stockwell et al. (2016). At the zigzag kiln, the on-line

sampling system was in operation for an ~4 hour sampling period, but unfortunately because of the heat of the kiln, the mAMS was only operational for a portion of the sampling period (~0.5 hours). The sampling period occurred between coal feeding cycles during a continuous firing period. The on-line $EF_{PM1}$ of the kiln was 1.823 g kg$^{-1}$ comprised of 52% sulfate, 26% BC, 16% OA, and 6% ammonium. The zigzag kiln is the only investigated emission source during NAMaSTE to have sulfate as the largest component of the PM$_1$ emissions and one of few to have more BC than OA. The filter samples were collected over

different time periods than the mAMS operation, but also found sulfate was a major component of the PM (Jayarathne et al., 2018). Additionally, the mass distributions shown in Fig. 3 indicate that sulfate aerosol with $d_{va}$ between 180 nm and 560 nm comprised 63% of the sulfate distribution and approximately 36% of the total measured PM$_1$ mass. The non-refractory species were found to have a mode $d_{va}$ of ~345 nm (Fig. 3).

Similar to the clamp kiln, PAHs were not observed above detection limits from the zigzag kiln. The PAH detection limit at the

zigzag kiln was estimated to be 42.3 ng m$^{-3}$ and the clamp kiln PAH detection limit was 90 ng m$^{-3}$. The observation of PAH concentrations below detection limits from the coal-fired kilns is unexpected because PAHs have previously been observed in coal emissions measured by an AMS in China and were proposed as tracer compounds for coal burning (Hu et al., 2013). PAH aerosol was also not readily observed at the zigzag kiln by the NAMaSTE filter-based measurements, but was observed from the clamp kiln with an EF of 18.7 mg kg$^{-1}$ (Jayarathne et al., 2018).

Although the two brick kilns had roughly similar $EF_{PM1}$ and size distributions, the two coal-fired kilns had major differences in efficiency and in the chemical composition of emissions. The zigzag kiln had lower OA emissions and markedly enhanced BC and sulfate EFs compared to the clamp kiln. Additionally, the mass spectral profiles of the brick kiln OA emissions indicate that significant compositional differences existed between the kiln emissions primarily because the zigzag kiln emissions contained nitrogen-containing organic compounds that were not present in the clamp kiln emissions (Goetz et al., 2018). The differences in

OA and BC between the two kilns are thought to be due to the enhanced combustion efficiency of the zigzag kiln (MCE = 0.994)





compared to the clamp kiln (MCE = 0.950) (Stockwell et al., 2016). Mass spectral data showing low $f_{60}$ (organic signal at m/z 60 ratioed to the total organic signal) indicates that emissions of levoglucosan, a biomass burning tracer compound, were limited at the clamp kiln compared to other biofuel sources suggesting that coal was the dominant fuel inside the kiln and wood burning was limited at the time sampling took place (Goetz et al., 2018;Jayarathne et al., 2018). Therefore, the difference in average

MCE between the kilns was likely because of kiln design rather than differences in fuel type. The role of kiln design on combustion efficiency was expected as the forced-draft system of the zigzag kiln is designed for enhanced fuel and production efficiency compared to the less-advanced clamp kiln. Alternatively, fuel quality can explain the differences in sulfate EF observed between the two fuels. Elemental analysis indicated that the zigzag kiln coal was composed of 1.28% sulfur and the clamp kiln was composed of 0.68% S (Stockwell et al., 2016). The larger sulfate EF at the zigzag kiln was therefore likely due to

the higher sulfur content of the coal used at the site.

Of the limited reports of brick-kiln emissions the results presented above agree well with what has previously been observed from coal-fired kilns. Weyant et al. (2014), conducted measurements at three South Asian zigzag kilns that operated with 100% coal and found an average $PM_{2.5}$ fuel-based EF of 0.93 g kg$^{-1}$ and an average $EF_{EC}$ of 0.43 g kg$^{-1}$. Assuming that the $PM_{2.5}$ observed by the Weyant et al. (2014) is roughly equivalent to $PM_1$ based on the observed mass distributions from this study, the

results from this study correspond well with previous zigzag kiln observations. Observations of biomass-fueled clamp kilns by Christian et al. (2010) in Mexico found an average $EF_{OC}$ of 0.18 g kg$^{-1}$ and an $EF_{EC}$ of 1.05 g kg$^{-1}$ produced under burning conditions with an average MCE of 0.968. The order of magnitude difference in $EF_{BC}$ from the clamp kiln in NAMaSTE compared to the $EF_{BC}$ from the biomass-fueled clamp kilns in Mexico combined with enhanced MCE at the Mexican clamp kilns, suggests that coal-burning kilns with lower efficiency could produce lower BC aerosol emissions compared to biofuel-

burning kilns on a per-unit-of-fuel basis though more samples of each type are needed. The use of coal for brick making in place of biofuels could therefore potentially reduce the climate impact of inefficient traditional brick firing operations. However, the role of other light absorbing aerosol emissions from brick kilns needs to be better quantified before fuel recommendations can be made for the mitigation of short-term climate forcers. For example, the clamp kiln emissions investigated in this study had strong ultraviolet absorption (Section 3.6) based on the aethalometer and PAX, which observed high AAE. This demonstrates that the

light absorbing properties of brick kiln emissions cannot be determined from BC quantification alone.

### 3.4 Crop residue burning

Emissions from the open burning of crop residues common in the IGP were investigated in NAMaSTE. In this work, segregated piles of mustard, grass, and wheat straw were burned and sampled in addition to a mixture of residues that included grass, wheat and rice straw, lentils, and mustard. The mixed residue was found to have an $EF_{PM1}$ of 3.44 g kg$^{-1}$ with compositional fractions of

0.77 OA, 0.12 BC, 0.10 chloride and nominal fractions of nitrate and sulfate (Fig. 1). Mustard and wheat residues were observed to have larger $EF_{PM1}$ at 4.18 g kg$^{-1}$ and 4.55 g kg$^{-1}$, respectively, with similar $PM_1$ composition (Fig. 1). Additionally, OA emissions from wheat straw burning had the largest variability of crop residues with a 10$^{th}$ percentile of 0.42 g kg$^{-1}$ and a 90$^{th}$ percentile of 18.78 g kg$^{-1}$. Grass burning was observed to have the lowest $EF_{PM1}$ of the tested crop residues (2.69 g kg$^{-1}$), but with an enhanced fraction of chloride emissions at 0.20. Black carbon, nitrate, chloride, and PAHs were all observed above the

background from all crop residue burns. Those EFs can be found in Figure 1 and Table S2.

The mass distributions for the crop-residue burns can be seen in Fig. 2. The majority of the OA mass from the burns was found in the accumulation mode (between 0.1 and 1 μm) and less than 15% of the OA mass was found below 100 nm $d_{va}$. Wheat burning OA was observed to have the lowest mode diameter of the investigated residues (240 nm $d_{va}$) and had the largest percentage of mass below 100nm $d_{va}$. Grass burning and the mixed residue OA emissions were observed to have mode $d_{va}$ at





300 nm and mustard burning OA had a mode $d_{va}$ of 400 nm. The differences in OA mass distributions are thought to be due to differences in fuel type and not due to differences in burn conditions since the average MCEs were roughly equivalent at ~0.955 for all of the residue burns except for mustard burning which had an MCE of 0.920 (Stockwell et al., 2016). Aside from OA, chloride was also observed to have distinguishable mass distributions from the crop residue burns and an ammonium distribution

was observed from grass burning. The chloride emissions were found to have mode $d_{va}$ that ranged between 130 nm to 175 nm and the ammonium distribution from grass burning emissions had a roughly equivalent mode $d_{va}$ to the chloride emissions. Similar chloride mass distributions with modes centered between 100 nm and 180 nm were also observed from sugarcane residue burning in Brazil (da Rocha et al., 2005). The significantly lower inorganic mode diameters compared to the mode diameters of the OA emissions indicate that the aerosol components were externally mixed. The non-refractory chloride measured by the

mAMS was estimated to be between 82% and 87% particle phase HCl. Additionally, for grass burning, HCl comprised up to 23% of the total non-refractory aerosol mass. Particle-phase chloride emissions were also observed by the filter-based measurements conducted by Jayarathne et al. (2018). Similar particle-phase chloride emission factors have also been observed with grass burning samples (0.31 g kg$^{-1}$) and agricultural waste samples (0.16 g kg$^{-1}$) from Africa (Keene et al., 2006). Conversely, gas-phase HCl emissions from crop residue burning were not observed above detection limits by Stockwell et al.

(2016). The preponderance of externally mixed, particle-phase chloride suggests condensation of HCl is occurring within the crop residue plumes and, unlike what was observed with garbage burning, the inorganic chlorine mass is mostly found in the particle phase. Organic chlorine, primarily in the form of chloromethane (CH$_3$Cl), however, was reported from the gas-phase measurements (Stockwell et al., 2016). The presence of both inorganic and organic chlorine emissions as large fractions of PM$_1$ mass from agricultural residues, combined with the large emission rates of aerosol produced from residue burning in parts of

South Asia (Pandey et al., 2014), suggests that crop residue burning, along with garbage burning, are major sources of atmospheric chloride in South Asia and globally.

### 3.5 Traditional mudstoves

We investigated aerosol emissions from three separate traditional mud cookstoves found in homes within the Tarai region of Nepal (Table 1). The stoves were operated with local biomass fuels common to South Asia; including hardwood sticks and

twigs, and dung. The hardwood sticks were from local sources and the dried dung logs were provided to the stove operators by the NAMaSTE team, as dung burning was not common in the particular location where sampling took place.

Emissions from the hardwood-fueled stove were sampled during an evening cooking cycle where lentils, rice and curry were cooked in a pressure cooker heated by the stove. The hardwood fuel was primarily Bakaino (*Melia azadarach*). The hardwood-fueled stove produced an EF$_{PM1}$ of 2.72 g kg$^{-1}$ over the ~1 hour burn period (Fig. 1). The PM$_1$ was comprised of 87% OA, 7.7%

BC, 4.5% chloride, and <1% sulfate (Fig. 1). The observed EF$_{PAH}$ was 12 mg kg$^{-1}$. Figure 2 indicates that hardwood burning had an OA mode $d_{va}$ of 200 nm and a chloride mode $d_{va}$ of 133 nm. Like the agricultural residue samples previously discussed, the chloride aerosol and OA appear to be externally mixed based on the differences in mass distributions. Nearly 84% of the OA mass was found in the accumulation mode. Similar size distributions have been observed elsewhere with oak and pine wood burning (Kleeman et al., 1999).

Emissions from a separate cookstove fueled with sticks and twigs of *Shorea robusta,* and ignited with plastic, were sampled during a morning cook cycle (Table 1). During the 1-hour long cook cycle lentils, roti, curry, and rice were prepared. It should be noted that the ignition and start-up phase of the stick burning started ~15 minute prior to sampling and therefore plastic burning was likely not part of the sampled emissions. The sticks and twig burning cookstove produced an EF$_{PM1}$ of 2.36 g kg$^{-1}$ that was composed of 76% OA, 22% BC, 1.5% chloride, and nominal fractions of sulfate and nitrate aerosol (Fig. 1). The EF$_{PAH}$ for stick





fueled cooking was 25 mg kg$^{-1}$, which was the largest PAH emission factor of the emission sources investigated in NAMaSTE. The large EF$_{PAH}$ from stick and twig burning is possibly explained by its larger bark content, which has previously been observed to emit high levels of PAHs (Weimer et al., 2008). Stick burning had the highest MCE of the tested cookstove fires (0.933; Stockwell et al. (2016)), and the more efficient burn conditions are thought to be responsible for the enhanced EF$_{BC}$ and reduced

EF$_{OA}$ compared to the hardwood-fueled stove. The stick-fueled stove did however have larger variability in PM$_1$ emissions compared to the hardwood-fueled stove, which was likely because of the differences in burn cycle due to the low density and inconsistency of stick fuel compared to hardwood logs (Fig. 1). Although differences in emission factors existed between the wood-biomass-fueled stoves, the two burns produced similar mass distributions. Like the hardwood-fuel cooking, the stick-fueled cooking was found to have an OA mode $d_{va}$ of ~190 nm and a chloride mode $d_{va}$ of 123 nm (Fig. 2).

The single-pot mudstove that was fueled with sticks and twigs was later separately fueled using cow dung logs. The dung logs were ignited with kerosene and the stove was operated for ~30 minutes and without cooking. The measured dung burning median EF$_{PM1}$ was 1.79 g kg$^{-1}$ and was composed of 76% OA, 15% chloride, 4.8% BC, 3.9% ammonium and less than 1% sulfate and nitrate (Fig. 1). Dung burning had the lowest EF$_{PAH}$ of the investigated biomass fuels with a median of 5 mg kg$^{-1}$, a 25$^{th}$ percentile of 2 mg kg$^{-1}$, and a 75$^{th}$ percentile of 6 mg kg$^{-1}$ (Fig. 1). Also it should be noted that the dung burning had the

lowest variability in OA emissions of the sampled biomass in NAMaSTE (Fig. 1). Compositionally, the dung-fueled cookstove emissions were distinct from the wood burning emission because of the lower BC emissions, the greatly enhanced chloride emissions, and because of the presence of ammonium in the aerosol. Significant chloride and ammonium emissions were also sampled by the off-line filter measurements and gas-phase HCl was not measured above detection limits indicating that particle-phase chloride was dominant with dung burning (Jayarathne et al., 2018;Stockwell et al., 2016). Assuming that all the

ammonium measured was a counter ion to the various anion species (SO$_4^{2-}$, NO$_3^-$, Cl$^-$), a predicted ammonium concentration that represents full anionic neutralization was calculated for the NAMaSTE dung-burning samples (RETS samples included). Based on the predicted values, anionic mass from dung burning ranged from 35% to 50% neutralized and the field samples were 45% neutralized. The presence of chloride as the dominant anion in the measured PM$_1$ combined with the lack of HCl observed in the gas-phase, suggests that there was chloride containing organic species present in the dung-burning aerosol, other non-refractory

chloride organic salts, or ionic potassium (K$^+$). Slow vaporizing K$^+$ was not observed by the mAMS, but was observed in filter samples to make up an average of 15% of the chloride mass emitted by dung burning (Jayarathne et al., 2018).

In addition to having unique emission factors, the dung-fired mudstove was found to have a unique mass distribution compared to wood burning. The OA mass distribution was observed to be bimodal with estimated $d_{va}$ modes at 150 nm and 270 nm, and a trough at ~200 nm (Fig. 2). Based on the 2-dimensional time series of the organic mass distribution found in Fig. 4 (top panel)

it's clear that the two distinct OA distributions materialize at different time periods during the dung-burning test. The two distinct distributions appear to correspond with the two identified modes from the average distribution in Fig. 2, with the larger mode $d_{va}$ occurring at the start of the burn and the smaller mode $d_{va}$ occurring shortly after ignition of the dung (Fig. 4). Additionally, during the ignition phase of the burn, the highest OA and BC EFs were observed and the MCE derived from the uncalibrated Picarro CRDS was at its highest point. Two minutes after ignition the inorganic components appeared in the emissions, the OA

and BC EFs decreased, and the OA distribution shifted to smaller sizes (Fig. 4). The MCE, however, did not appear to follow the same abrupt trend and remained constant with a relative value of ~0.97, suggesting that the dung remained in a flaming phase. Because the MCE did not follow the same trend as the aerosol, it is thought that the kerosene that was used to ignite the dung was responsible for the larger mode diameter of the OA distribution and for the absence of inorganic mass at the start of the burn. However, without further samples, it is unclear if the same trends in size and mass would occur with different starter fuels.





Although we did not sample other exclusively dung-fueled cookstoves in the field, we did sample cooking with a 2-pot traditional mudstove that was co-fired with dung and hardwood, and started with hardwood (Table 1). The 2-pot stove was used to cook rice, lentils, and curry during the evening cook cycle at a village restaurant. The co-fired stove did not show the same bimodal mass distribution as the single- pot dung-fueled stove and was observed to have a unimodal distribution that was most

similar to the hardwood-fueled cookstove. Unlike hardwood burning, the co-fired mass distribution was found to have large fractions of ammonium and chloride aerosol (Fig. 2). Organic aerosol emitted from the co-fired stove emissions was estimated to have a mode diameter of 206 nm. Inorganic aerosol was estimated to have a mode diameter of ~125 nm. Again, like the other cookstove emissions the differences between the organic and inorganic modes suggest that the aerosol components were externally mixed. Although differences in mode diameter existed between the aerosol components, the non-refractory

distribution was fairly narrow and 67% of the mass was found between 100 nm and 320 nm. Because the authors are not aware of other studies that have characterized the aerosol size distributions of dung-burning emissions in the field, we cannot comment on the universality of the distributions observed from the NAMaSTE samples. However, under more dilute conditions (~1:45) produced in a lab, Venkataraman and Rao (2001) found a mass median aerodynamic diameter of dung-fired emissions between 600-780 nm. It is important to note, that in NAMaSTE an effort was made to sample well mixed emissions from inside the

building in which sampling took place. If our well-mixed assumption holds true then the mass distributions observed in this study are more representative of residential exposure.

The dung and hardwood-fueled, two-pot traditional mudstove was found to have enhanced aerosol emission factors compared to the tested single-pot stoves. With an average MCE of 0.912 (Stockwell et al., 2016), the 2-pot stove was observed to have a $EF_{PM1}$ of 4.10 g kg$^{-1}$. The $PM_1$ was composed of 81% OA, 12% chloride, 3.9% BC, 2.6% ammonium, and nominal fractions of

sulfate and nitrate (Fig. 1). The co-fired cookstove had an enhanced $EF_{PAH}$ compared to both single-pot hardwood and dung burning at 20 mg kg$^{-1}$. The general increase in aerosol emissions between the dung-fired single-pot stove and the co-fired two-pot stove was unexpected because few differences between the two sources were observed by the filter-based measurements of Jayarathne et al. (2018) or the gas-phase measurements of Stockwell et al. (2016).

### 3.6 BrC absorption and AAE

Emission factors of absorption due to BrC ($EF_{BrC,abs}$) derived from the scattering and filter-loading  corrected aethalometer measurements at 370 nm can be found in Figure 5a. Absorption coefficients measured at 880 nm and 370 nm and given as emission factors can be found in Table S5 in the supporting information. Generally, BrC absorption at 370 nm was observed from all of the investigated emission sources with the exception of the idling motorcycles, the zig-zag kiln, and Mix 1 garbage burning, in which BrC was not observed above the background. The presence of light-absorbing organic aerosol from the

biomass burning samples was expected as BrC has primarily been attributed to biomass burning (Saleh et al., 2014). Agricultural residue burning $EF_{BrC,abs}$ ranged from 12.0-25.2 m$^2$ kg$^{-1}$, with the largest observed $EF_{BrC,abs}$ from wheat and mustard burning (Fig. 5a). The wood-fueled cookstove BrC emissions were consistent with an average $EF_{BrC,abs}$ of 15.2 m$^2$ kg$^{-1}$ from the field-tested stoves. BrC from the field-tested, dung-fueled cookstoves was found to be 14.8 m$^2$ kg$^{-1}$ for the single-pot stove emissions and 10.8 m$^2$ kg$^{-1}$ for the two-pot stove. With the exception of Mix 1, open garbage-burning emissions were observed to contain

$EF_{BrC,abs}$ equivalent to the biomass burning emissions with EFs ranging from 9.1-19.5 m$^2$ kg$^{-1}$. Another unexpected source of BrC was the diesel irrigation pumps. Enhanced light absorption at 370 nm has predominantly been associated with biomass burning aerosol and not primary diesel emissions (Olson et al., 2015). Pump 1, the older pump with enhanced OA emissions, was found to have an $EF_{BrC,abs}$ equivalent to the biomass burning sources at 12.6 m$^2$ kg$^{-1}$ (Figure 5a). Pump 2 emissions were found to have significantly lower level of BrC (2.4 m$^2$ kg$^{-1}$). Similarly low $EF_{BrC,abs}$ was observed from the clamp kiln (3.6 m$^2$ kg$^{-1}$). Light




absorbing OA from coal burning has also been observed by other studies (Bond et al., 2002;Olson et al., 2015;Sun et al., 2012) and at both brick kilns by the NAMaSTE PAX measurements at 405 nm (Stockwell et al., 2016).

With the exception of the zig-zag kiln emissions, the observed $EF_{BrC,abs}$ measured with the PAX at 405 nm were an average of 50% lower than the 370 nm AE33 results from this work. The consistently enhanced $EF_{BrC,abs}$ from this work that are partly

because the 40 nm difference in wavelength used to measure absorption generated differing results. Thus, the 370 nm BrC results from this work were converted to 405 nm using the AE33 AAE to make comparisons to Stockwell et al. (2016). For the conversion it was assumed that AAE is independent of wavelength and that the differences in absorption at longer wavelengths are minor (i.e. 870 nm versus 880 nm). Based on the converted results there was a 25% increase in the average agreement between the PAX and AE33 results (Fig 5a). Additionally, for crop residue burning and mustard burning, there was a >90%

agreement between the 405 nm $EF_{BrC,abs}$. It should also be noted that our indirect 405 nm results dramatically decreased the $EF_{BrC,abs}$ for garbage-burning and the irrigation pump emissions, which produces some non-positive BrC values for the mixed refuse 2, chip bags, and pump 2. The observation of excess BrC absorption at 370 nm compared to 405 nm is expected to be larger at higher AAE and confirms the general shape of the cross-section assumed in the optical model.

Absorption Ångström exponents from the investigated emission sources were generally divided into three groups with low AAE

observed with the open garbage-burning, irrigation pump, and zig-zag kiln emissions (0.8-1.4), moderate AAE observed from the agricultural residue burning, wood-fueled cookstoves, and the mixed-fuel stove emissions (2.1-2.9), and high AAE from the dung-fueled cookstove and the clamp kiln emissions (3.7-4.1) (Fig. 5b). Generally, the AAE results from Stockwell et al. (2016), followed similar trends, but with significantly larger AAE values associated with the zig-zag kiln emissions and wheat straw burning (Fig. 5b). Evidence of outlying zig-zag kiln $EF_{BrC,abs}$ and AAE compared to the PAX combined with an AE33 AAE

significantly <1 suggests a possible sampling artifact associated with the 370 nm zig-zag kiln results. For example, the outlying zigzag kiln results could be due to an under estimation of scattering associated with the zigzag kiln emissions. With regard to the garbage burning and irrigation pump optical results, the AAE values >1 provide further evidence for the presence of some BrC or coated BC (Pokhrel et al., 2017).

### 3.7 OC/BC

The large instrument suite used in NAMaSTE provided unique insight into the chemical composition of emissions from prevalent emissions sources found in South Asia, but also generated complex and sometimes diverse results. As discussed briefly with the above results, some differences in aerosol EFs were observed between the two on-line techniques used in NAMaSTE. In addition, the on-line EF sometimes differed from EF results found by off-line analysis. The differences observed between the instrumentation are thought to be due to the inherent differences between detection methods including cut size (i.e. $PM_1$ vs

$PM_{2.5}$), BC or EC operational definitions (Hitzenberger et al., 2006;Watson and Chow, 2002), and mass quantification of refractory or semi-refractory organic components. Partly due to sampling methodology (i.e. controlled dilution vs. ambient dilution), and also the lack of exact temporal and spatial overlap in dynamic sources. The lack of exact overlap between the measurements actually represents a beneficial increase in the total sampling, but nevertheless, other contributions to differences can be explored. Here we revisit and summarize the mass ratio of organic carbon to black carbon (OC/BC) from the emission

sources studied in NAMaSTE. The mass ratio provides an internally-consistent parameter to assess the aerosol composition between emission sources and also offers a metric to make comparisons of aerosol composition across the NAMaSTE instrumentation and relevant results in the literature. For this analysis AMS OA data has been converted to OC using OM/OC ratios as described previously in section 2.3. The OC to BC mass ratio of the field-tested emission sources can be found in Fig. 6 and are displayed in rank order. Additionally, the marker for each emission source in Fig. 6 is colored by the average MCE from





Stockwell et al. (2016). Generally, the lowest OC/BC values appear to correspond with the highest observed MCEs. The trend between OC/BC and MCE matches observations from biomass burning emissions that have taken place in the field (Kondo et al., 2011) and in the lab (Christian et al., 2003).

The largest on-line OC/BC was observed at the coal-fired clamp brick kiln with a value of 52.4 (Fig. 6). Conversely, the OC/BC observed at the zigzag brick kiln was ~0.5 and the lower zigzag kiln percentiles ranged to below 0.1. Similar results were observed with other South Asian coal-fired zigzag kilns by off-line filter-based measurements with OC/EC ratios ranging from 0-0.29 (Weyant et al., 2014). Clamp kilns investigated in Mexico saw similarly low OC/EC emissions with an average of 0.16 (Christian et al., 2010). The lower OC/EC observed with clamp kilns in Mexico compared to the clamp kiln investigated in this study could be due to fuel type as discussed in Section 3.3.

The OC/BC ratios did not follow any specific trend based on source type, although the agricultural residue-burning samples were grouped with ratios between 3.7 and 4.5 (Fig 6). Literature based results provide an estimated range of 1.8-58 for OC/EC for crop residue burning (Andreae and Merlet, 2001;Cao et al., 2008;Hays et al., 2005;Liu et al., 2016;Sahai et al., 2007). Additionally, the off-line filter based measurements of Jayarathne et al. (2017) are in the range of literature values with an estimated OC/EC of 6.44 (Figure 6).

Open garbage burning produced varied results with Mix 1 and plastic-burning emissions containing OC/BC >5.5 and Mix 2 and metalized plastic chip bags containing OC/BC ≤1 (Fig 6), which follows the trend of low OC/BC from high MCE sources and high OC/BC from lower MCE sources. Based on the EFs, the differences between Mix 1 and Mix 2 are largely due to enhanced BC emissions observed from Mix 2. The differences between the plastic-burning samples were primarily due to enhanced OA emissions from mixed plastic compared to the chip bags (Fig. 1). Based on the plastic-burning results, it's possible that Mix 1 was composed of a larger percentage of mixed plastic compared to Mix 2, although differences in burn conditions are likely also a factor. Christian et al. (2010) also observed OC/EC >1 (2.3-28.5) from open garbage burning in Mexico. The combined OC/BC range from this study and Christian et al. (2010) suggests that considerable variability in garbage-burning emissions occurs over a range of MCE and moisture.

The diesel-powered irrigation pumps were observed to have the most consistently low OC/BC of the source types but emitted noticeably different ratios (Pump 1 OC/BC = 1.88; Pump 2 OC/BC = 0.15). As discussed in Section 3.2 emissions from Pump 2, the newer more efficient pump, had considerably reduced OA EFs compared to the older Pump 1 (Fig. 1). Here the reduced OA associated with Pump 2 is responsible for the lower observed OC/BC. Other reports of OC/BC from diesel irrigation pumps do not exist outside of NAMaSTE, but our observed OC/BC are close to or within the lower range of observations from diesel-powered military generators which emitted an estimated OM/BC range 0.23-6.25 (Zhu et al., 2009). Additionally, OC/EC observed from U.S. diesel vehicle primary emissions under dilute conditions (>1:8 dilution) were found to be less than 1 (May et al., 2014) in agreement with the on-line results for the newer pump in this work. Although we only sampled two irrigation pumps in NAMaSTE, the results indicate that the aerosol emissions are similar to what has been observed from other diesel combustion sources by other studies. However, a better understanding of inter-pump variability and the effects of aging and maintenance are needed before emission factors from other more common (and likely more controlled) emission sources can used as a supplement for field-tested diesel-powered irrigation pump aerosol emissions.

The wood-fueled traditional mudstoves were observed to have emissions with OC/BC close to what was observed from the crop-residue burns and the filter-based measurements of Jayarathne et al. (2017). The NAMaSTE filter-based EF measurements taken from the same sources at different times were within ±30% of the on-line measured values (Fig 6). The OC/BC from the hardwood-fueled stove was found to be 8.1 and the stick-burning stove emissions had an OC/BC of 2.5. The OC/BC observed from the wood-fired traditional stoves in Nepal are within the estimated OC/EC range of 1.22-11.5 observed from similar wood-





fired stoves investigated in Guatemala (Roden et al., 2006) and Mexico (5.2 ± 3.6, Christian et al. (2010)). Additionally, OC/BC of the hardwood-fired cookstoves tested at RETS ranged from 0.34-4.72 from traditional stoves with and without chimneys, natural draft improved stoves, a Bhuse Chulo, and a 3-stone fire, with the largest OA/BC observed from the 3-stone fire. Another study that investigated South Asia residential biofuels found an average OC/EC of 0.5 for low burn rate fuelwood and an OC/EC

of 3.8 for high burn rate fuelwood (Venkataraman et al., 2005). The results suggest that the NAMaSTE emission results provide important additional sampling to the aerosol composition results that have been reported by other studies with larger sample numbers, but far less chemical detail.

The field-tested dung-fueled mudstoves were found to have some of the largest OC/BC observed in the study. The 100% dung-fueled test was observed to have an OC/BC of 11.8 and the co-fired dung and hardwood test has an OC/BC of 15.1 (Fig 6). The

filter-based measurements by Jayarathne et al. (2017) were found to have a similar trend to the on-line measurements and the OC/EC from both tests were within ±70% of the online measurements (Fig 6). Other studies that have investigated dung-burning emissions in the lab have observed OC/EC ratios of greater than 20 (Sheesley et al., 2003; Venkataraman et al., 2005). Therefore, it's possible that the consistently lower OC/BC from the field-tested NAMaSTE results are more representative of authentic dung-fueled cookstove emissions.

With many of the aerosol EF results it has been discussed how differences exist between the two sets of on-line EFs from this work and also with the off-line results in the companion paper by Jayarathne et al. (2018). As previously discussed many of the large differences in $EF_{OC}$ and $EF_{BC}$ could be due to detection method, sampling overlap, or differences in dilution. As seen in Fig. 6, there is some agreement in OC/BC between the on-line and off-line measurements and in particular there is agreement with many of the biomass burning emission sources. However, there are significant differences between online and offline

OC/BC of the sources with high MCE (i.e. irrigation pumps and chips bag burning). Additionally, it should be noted that elemental carbon wasn't observed above detection limits from the brick kilns or from garbage mix 2 by Jayarathne et al. (2018), and a low $EF_{EC}$ could inflate the OC/EC. Figure 7 compares the on-line $EF_{BC}$ to each other and the offline $EF_{EC}$ from the NAMaSTE tested sources in which results from the same tests were available. The figure indicates that there was significant scatter between the online $EF_{BC}$ results and the off-line $EF_{EC}$ results. Many of the high MCE emission sources (i.e. irrigation

pumps, garbage burning, plastic burning, zig-zag kiln) were found to have lower $EF_{EC}$ compared to the $EF_{BC}$ results or EC was not detected by the off-line filter based methods. A low EF for EC could help explain why OC/EC was higher than OC/BC for these sources (Fig 6). Additionally, although the sample size is limited, emission sources with AAE values significantly greater than 1 generally had larger off-line measured $EF_{EC}$ compared to the on-line $EF_{BC}$ (Fig. 7). Similar results were also observed when comparing the off-line $EF_{EC}$ to on-line PAX measurements by Stockwell et al. (2016) (Fig 7). The results suggest that there

could be two processes, in addition to sample timing that contributed to the large differences in off-line and on-line BC EFs and were partially responsible for inconsistent OC/BC: (1) black carbon mass from high MCE sources was quantified differently between off-line and on-line detection methods, and (2) the presence of light-absorbing organic carbon impacted BC (and/or EC) detection. Similar processes have been observed with ambient measurements that compared thermal-optical transmittance derived EC to BC from an aethalometer, where urban haze events were found to have larger aethalometer measured BC

concentrations and biomass burning events were observed to have larger off-line measured EC concentrations (Jeong et al., 2004). In NAMaSTE the online results were better correlated with an $r^2$ of 0.34, but with poor agreement between the dung and plastic burning results from the two instruments (Fig 7). The general agreement between the PAX and AE33 indicates that the manufacturer-selected mass absorption cross section of the AE33 at 880 nm (7.77 $m^2$ $g^{-1}$; (Drinovec et al., 2015)) was likely applied correctly for most sources, but may have been incorrectly assumed for dung and plastic burning In summary, ultimately,

the source of variability between the various detection methods cannot be fully elucidated in this work and further experimental





controls that were not available in the field are needed to fully characterize these differences. More than likely, all the techniques provided information that is worthy of incorporating into evolving literature averages.

## 4 Summary

On-line $PM_1$ measurements of emissions from prevalent but under-characterized emission sources in South Asia using a mAMS and aethalometer were conducted as part of the NAMaSTE field campaign. With controlled dilution sampling, fuel-based emission factors of major aerosol species were derived from the time-resolved measurements of the field-tested emission sources. Additionally, mAMS-measured average mass size distributions were generated from each emission source. The field-tested emission sources included traditional mudstoves, agricultural residue burning, brick kilns, open garbage burning, ground water pumps used for irrigation, and idling motorcycles. Open garbage burning, a globally important but poorly understood emission source, was found to have some the largest and most variable $PM_1$ emissions of the sources investigated in NAMaSTE. Like previous open garbage burning observations, particle-phase chloride was observed from the combustion of PVC plastic, but based on other complementary measurements chlorine mass was primarily in the gas-phase as HCl. Diesel-powered irrigation pumps were also observed to have large $PM_1$ emission factors compared to other investigated sources. The on-line measurements indicate that the two sampled ground water irrigation pumps produced similar OA size distributions (mode $d_{va} \approx 80$ nm), but produced significantly different OA emission factors. Differences in efficiency due to age or model are thought to be responsible for the different $EF_{OA}$ from the pumps. The OA size distributions obtained from the motorcycle emissions agreed well with what has been observed from emissions of other Asian motorcycles. The two mainly coal-fired brick making kilns were observed to have similar $PM_1$ emissions on a per mass of fuel basis but with some differences in composition that are thought to be due to differences in design. The traditional and less efficient clamp kiln had a larger $EF_{OA}$ compared to more efficient zigzag kiln and the largest AAE of the investigated emission sources (~4). The zigzag kiln was observed to have a larger sulfate EF due to the higher sulfur content of the coal used for firing, and the organic aerosol was found to contain nitrogen-containing species. Polycyclic aromatic hydrocarbons were not observed above the mAMS detection limits from either of the coal-fired brick kilns. Crop residue burning EFs were found to be within the range of other crop residue experiments found in the literature. Interestingly, chlorine aerosol emissions externally mixed from organic aerosol were observed from the crop residue experiments and unlike observations from open garbage burning, a significant fraction of chloride mass was found in the particle phase. Aerosol emissions from traditional mudstoves used for cooking and fueled with hardwoods and dung were investigated in the field. For all of the cookstove experiments, organic aerosol was the dominant aerosol component in the emission with OC/BC ranging between 2.4 and 15.1. Like crop residue burning, chloride aerosol was observed from all of the cooking experiments and was externally mixed from the organic aerosol based on the size distribution data. Ammonium emissions were observed with dung burning suggesting that the emissions were neutralized to some extent. Additionally, the largest $EF_{PAH}$ was observed from the single-pot mudstove fueled with stick and twigs, which is thought to be due to high bark content.

In addition to examining size distributions and speciated emission factors, aerosol optical properties and mass ratios of black carbon and organic aerosol were summarized for the investigated emission sources. The agricultural residue burning, wood and dung fueled cooking, and the clamp kiln were all observed to have AAE values > 2. The clamp kiln and dung-burning emissions were observed to have the highest wavelength dependence (AAE) and a similar trend was observed in the aerosol optical analysis in a companion paper by Stockwell et al. (2016). Ratios of organic carbon to black carbon were examined to make comparisons of composition between the emission sources, the filter-based aerosol measurements in the companion paper Jayarathne et al. (2018), and literature values. The OC/BC measurements made in this work corresponded well with other studies



that have investigated similar sources indicating that the on-line emission factors presented in this work both support and supplement previous results. Specifically, the aerosol size and composition results from this work have added important results to the literature for some prevalent, but under-characterized emission sources found in South Asia. The NAMaSTE results as a whole have expanded the body of knowledge about South Asia combustion sources and provide key results that will help

constrain uncertainty in emission inventories and indoor exposure models.

**5 Acknowledgements**

The authors would like to thank the logistics and support team in Nepal, which included numerous personnel of ICIMOD, MinErgy Pvt. Ltd., and RETS. We would like to acknowledge the proficiency and expertise of S.B. Dangol and the MinErgy team who were integral to finding and gaining access to field sites for this study. A special thanks to K. Sherpa, and Nawraj, who

were both central parts of the field team and without whom this work would not be possible and possibly less enjoyable. Finally, we thank the villagers of Nawalparasi for their generosity and hospitality. This project was funded by the National Science Foundation grant AGS 1461458. E. A. S. and T. J. were supported by NSF grant number AGS 1351616, R. J. Y. and C. E. S. were supported by AGS 1349967, and R. J. Y. was also supported by NASA Earth Science Division Award NNX14AP45G.

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





**Table 1.** Fuel type, location of sampling, and number of samples from tested emission sources.

| Emission Source | Source Type | Location | Fuel Type[a] | Samples[b] |
|---|---|---|---|---|
| **Brick Kilns** | Forced-draft Zigzag | Dhading District | Coal (Bagasse) | 1 |
| | Batch Style Clamp | Kavre District | Coal (Sawdust and HW) | 1 |
| **Motorcycles** | 4-stroke – idling | Kathmandu Valley | Gasoline | 4 |
| **Irrigation Pumpset** | Groundwater | Tarai village | Diesel | 2 |
| **Cookstoves** | 1-Pot Mudstove | Tarai village, RETS | HW, Sticks, Dung | 3(3) |
| | 2-Pot Mudstove | Tarai village | HW and Dung | 1 |
| | Chimney Stove | RETS | HW, Sticks, Dung | (3) |
| | Natural-draft Stove | RETS | HW, Dung | (2) |
| | Forced-draft Stove | RETS | Charcoal, HW | (2) |
| | Biolite Stove | RETS | Charcoal Briquettes | (1) |
| | Bhuse Chulo | RETS | Sawdust | (1) |
| | Biogas | RETS | Biogas | (1) |
| | 3-stone Heating Fire | Tarai village, RETS | HW, Sticks, Dung | (3) |
| **Open Garbage Burning** | Mixed Garbage | Kathmandu, Tarai village | | 2 |
| | Metalized Plastic | Kathmandu Valley | | 1 |
| | Plastic | Kathmandu Valley | | 1 |
| **Agricultural Residue Burning** | Mixed[c] | Tarai village | | 1 |
| | Wheat straw | Tarai village | | 1 |
| | Grass | Tarai village | | 1 |
| | Mustard | Tarai village | | 1 |

Note: 'RETS' is the Nepal Renewable Energy Test Station cookstove lab.
a. primary fuel (secondary or starter fuel)
b. number of field samples (number of RETS lab samples)
c. rice, wheat, mustard, lentil, grasses

**Table 2.** $f_{44}$, O:C, and OA:OC of the field-tested emission sources.

| Emission Source | $f_{44}$ | O:C[a] | OA:OC[b] | % Uncertainty |
|---|---|---|---|---|
| Clamp Brick Kiln | 0.008 | 0.114 | 1.324 | 6.2 |
| Zigzag Brick Kiln | 0.142 | | | 11.8 |
| Mixed Garbage | 0.015 | 0.144 | 1.361 | 5.4 |
| Metalized Plastic | 0.020 | 0.167 | 1.391 | 6.2 |
| Mixed Plastic | 0.019 | 0.162 | 1.384 | 5.3 |
| Motorcycles | 0.003 | 0.092 | 1.296 | 2.7 |
| Irrigation Pumps | 0.009 | 0.118 | 1.329 | 4.8 |
| Hardwood[c] | 0.024 | 0.182 | 1.409 | 1.3 |
| Sticks and Twigs[c] | 0.023 | 0.177 | 1.403 | 5.1 |
| Dung[c] | 0.013 | 0.137 | 1.352 | 4.0 |
| Dung and Hardwood[d] | 0.014 | 0.139 | 1.355 | 6.2 |
| Agricultural Residues[e] | 0.025 | 0.186 | 1.415 | 4.6 |

a. Based on linear relationship to $f_{44}$ from Canagaratna et al. (2015) (O:C = $4.31(f_{44}) + 0.079$)
b. Based on linear relationship to O:C from Aiken et al. (2008) (OA:OC = $1.260(O:C) + 1.180$)
c. Fuel used in single-pot traditional mudstove
d. Fuel used in two-pot traditional mudstove
e. Combined values for mustard, grass, wheat, and mixed residue piles (rice, wheat, mustard, lentil, and grasses)



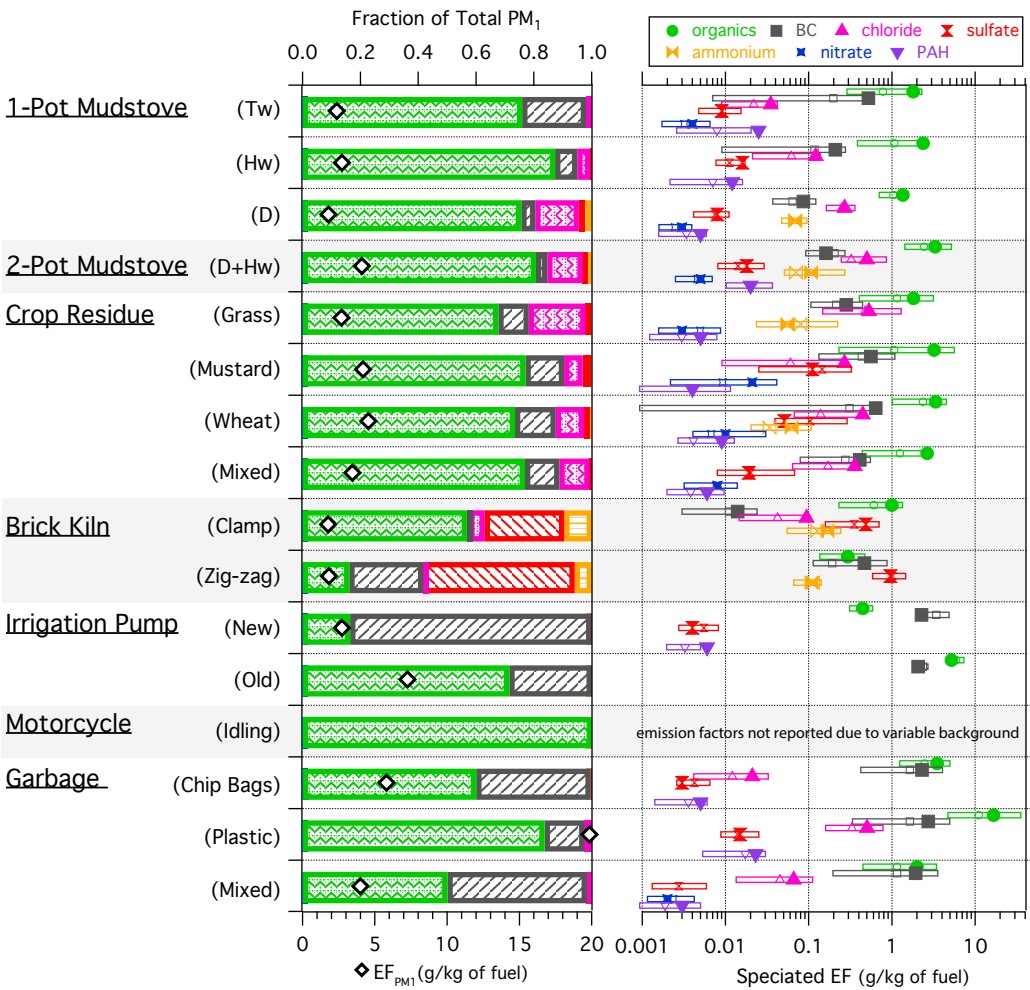

**Figure 1.** Summary of $PM_1$ fuel based emission factors and compositional fractions for the investigated emission sources. The colors of the horizontal bar chart correspond with the species colors designated by markers in the speciated emission factor panel. In the right panel the closed markers represent the test integrated average speciated emission factor, the open markers represent the median speciated emission factor and the bars represent the 25th and 75th percentile speciated emission factors for the combined observations from each emission source.





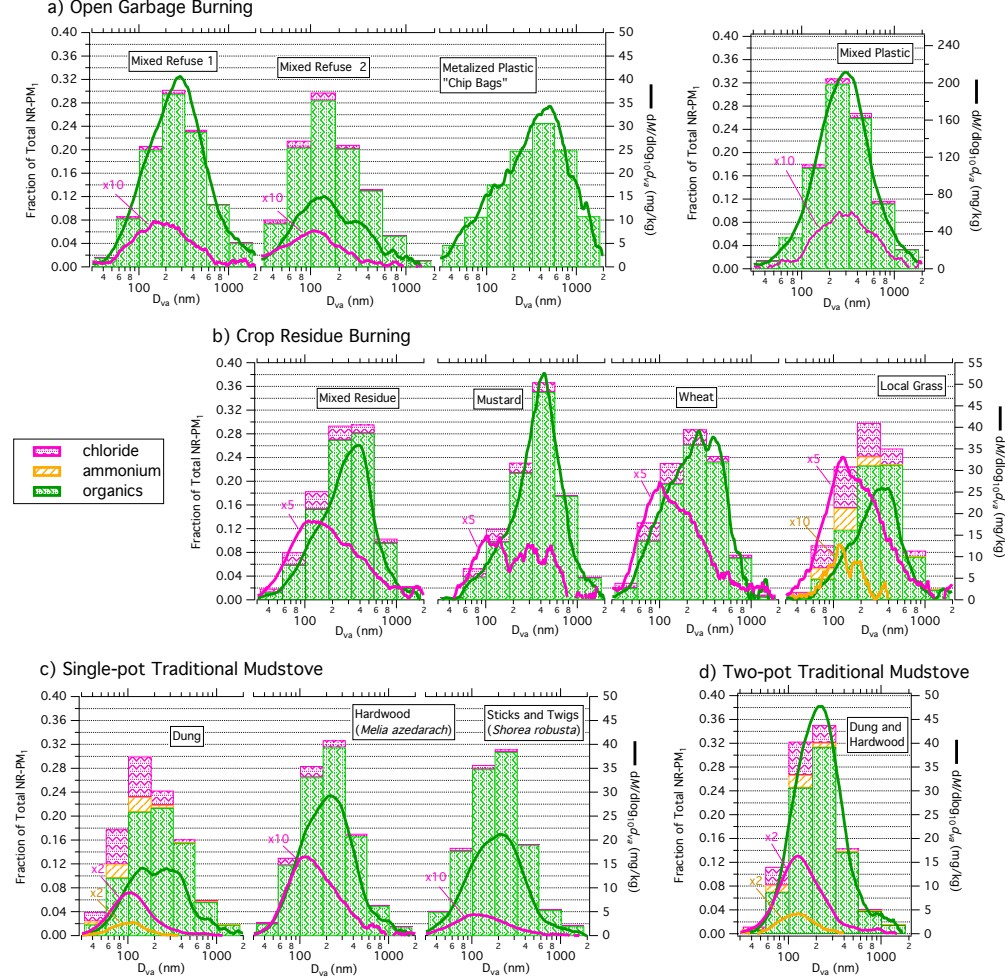

**Figure 2.** Size-resolved emission factors of (a) open garbage burning, (b) agricultural residue burning, and (c and d) cooking with a traditional mudstove. Continuous species specific mass size distributions normalized by the test integrated emission factor are shown as solid lines. Cumulative binned mass size distributions normalized by the total non-refractory submicron aerosol mass (NR-PM$_1$) are shown as stacked bars. The distributions bins are between 32, 56, 100, 180, 320, 560, 1000, 1800 nm. All sizes are shown as vacuum aerodynamic diameters.





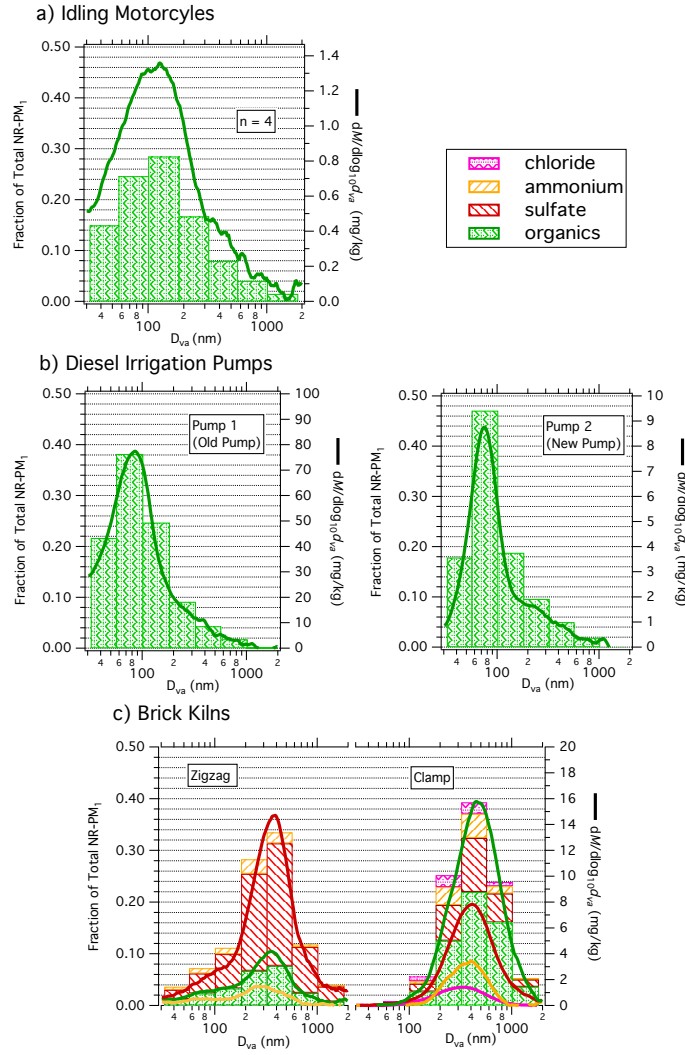

**Figure 3.** Size-resolved emission factors of (a) idling motorcycles, (b) diesel powered irrigation pumps, and (c) brick kilns. Continuous species specific mass size distributions normalized by the test integrated emission factor are shown as solid lines. Cumulative binned mass size distributions normalized by the total non-refractory submicron aerosol mass (NR-PM$_1$) are shown as stacked bars. The distributions bins are between 32, 56, 100, 180, 320, 560, 1000, 1800 nm. All sizes are shown as vacuum aerodynamic diameters.



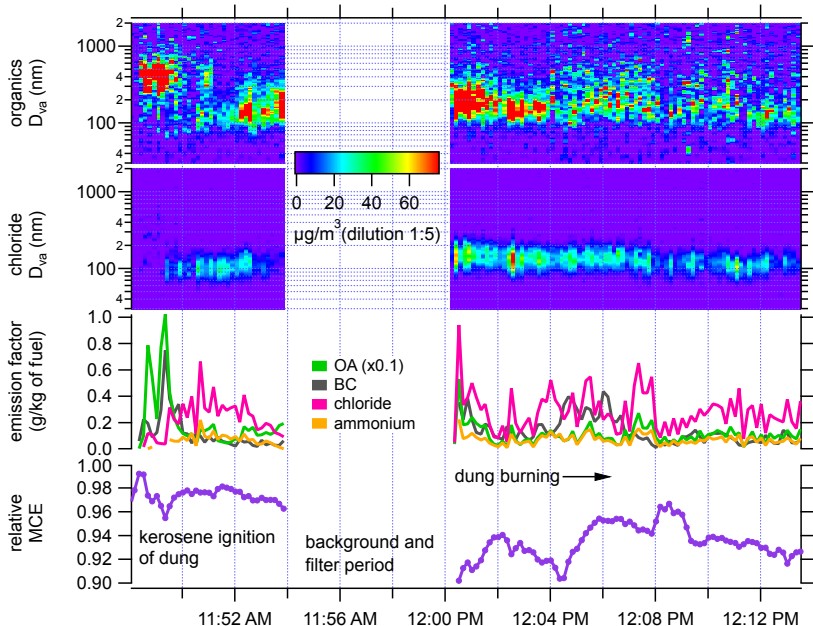

**Figure 4.** Time Series of the modified combustion efficiency (bottom panel), aerosol emission factors, and chloride and organic mass size distributions (top panels) for the dung burning emissions from a 1-pot traditional mudstove. The 2-d time distributions are colored by the size-resolved dilute concentration as indicated by the color ramp.



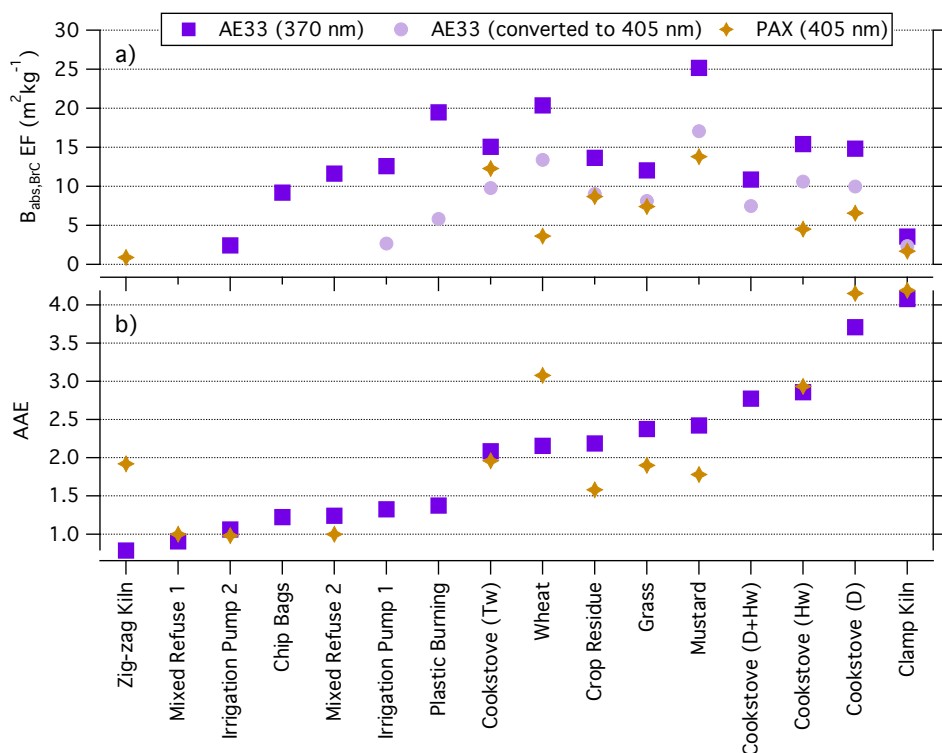

**Figure 5.** (a) Emission factor of aerosol absorption due to brown carbon detected at 370 nm by the AE33, converted to 405 nm, and at 405 nm by the PAX system (Stockwell et al., 2016). (b) Absorption Ångström exponent (AAE) for the investigated emission sources. Cookstove fuel types are hardwood (Hw), dung (D), and sticks and twigs (Tw).



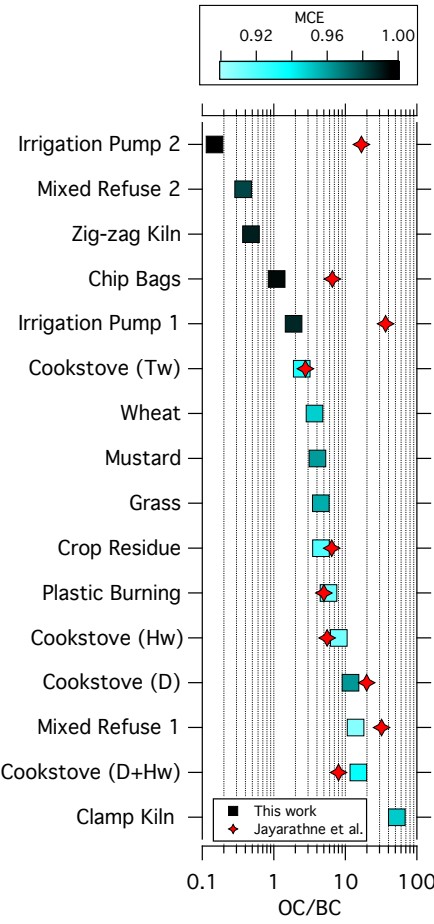

**Figure 6.** The organic carbon (OC) to black carbon ratio (g/g) of the investigated emission sources. The markers are colored by average MCE values from Stockwell et al. (2016), light blue indicates MCE of ~0.90 and black indicate a MCE close to 1. Cookstove fuel designated as D for dung burning, Hw for hardwood burning, and Tw for sticks and twigs burning.



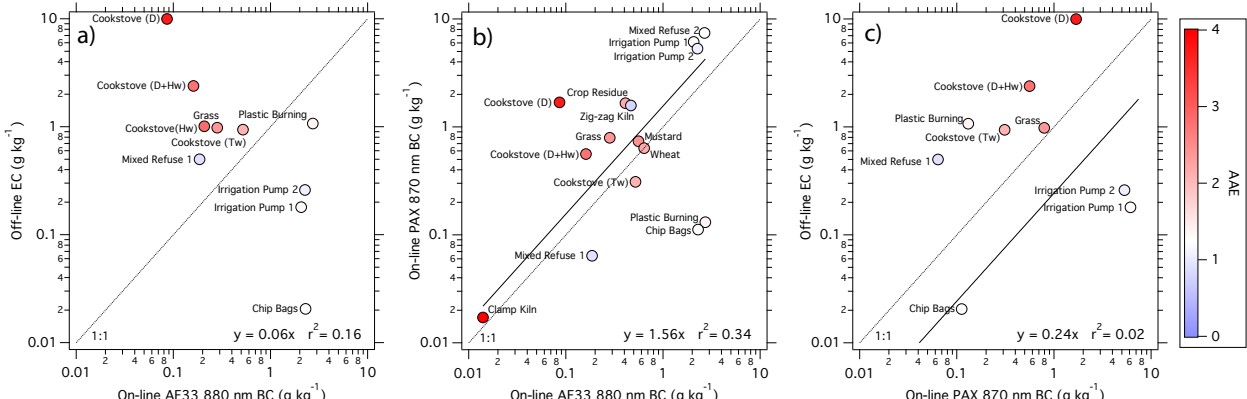

**Figure 7.** Scatter plots of (a) off-line thermal-optical measured elemental carbon emission factors from Jayarathne et al. (2017), (b) on-line PAX 870 nm measured black carbon emission factors from Stockwell et al. (2016) versus on-line AE33 aethalometer 880nm measured black carbon emission factors in units of g kg of fuel[-1]. Panel (c) provides a scatter plot of the off-line EC measurements versus the on-line PAX 870 nm measurements. Linear regression curves are displayed as solid lines with the resulting equation and correlation coefficient displayed in each respective panel. Markers are colored by the AAE for each emission source. Cookstove fuel types are hardwood (Hw), dung (D), and sticks and twigs (Tw). 1:1 lines (dotted) are shown in each plot for clarification.