# Peer review of "Speciated On-line PM1 from South Asian Combustion Sources: *Part I, Fuel-based Emission Factors and Size Distributions"

_Atmospheric Chemistry and Physics, 2018_

## Referee Comment (RC1) · Anonymous Referee #1 · 29 Apr 2018

This paper reports the AMS, Aethalometer and PAX data from the NAMASTE experiments, deriving emission factors for various small-scale pollution sources in Nepal. There are already other papers arising from this experiment, but presenting data from different instruments, so this work stands on its own right. The work is methodically presented and thorough, and is well within ACP's remit as there is relatively little authentic data on these sources, which are prominent in developing countries. I recommend publication after minor corrections.

Comments:

Because a technical citation for the mini AMS is not given, more of the specific technical

details should be listed, in particular the new data acquisition system. Is it the same ADQ system as the new system on newer models of AMS?

How confident that the Li-Corr factory calibration is still valid? Technical data to back up this assumption should be given, especially because this was operated under very different conditions to the laboratory. Was a post-calibration performed? Were any low pressure calibrations done in the laboratory? Ideally, an uncertainty estimate should be attached to this.

The uncertainty estimate attached to the OM:OC estimation based on f44 should be qualified better. Is this precision or accuracy? I would expect the accuracy to be questionable; it is known the relationship between OM:OC and f44 is both instrument and aerosol type specific and there are some types here that are new to the AMS. I would add additional caveats to this effect.

When describing the Lungdren plots, the authors fail to draw the distinction between continuum/transition aerodynamic used by impactors and vacuum aerodynamic diameters used by the AMS. Which is strange, considering the corresponding author was the first author on the definitive paper on this topic.

A moderate AAE does not necessarily imply the presence of BrC; BC particles with non-absorbing coatings can also exhibit this, depending on the primary spherule size (Liu et al., Geophys. Res. Lett., 42, 613-619, 10.1002/2014GL062443, 2015). While the authors touch on this, it isn't very clear.

Figure 1: The acronyms next to the mudstoves should be explained in the caption.

Figure 4: Reporting the local time of the measurement is not meaningful. It would be better to use 'time since ignition' as the x-axis.

---

## Referee Comment (RC2) · G. McFiggans (Referee) · 27 Jul 2018

This is a well-written paper describing a careful, comprehensive and challenging set of emissions measurements of significance throughout the developing world. The introduction provides convincing motivation for the NAMaSTE experiment as a whole and specifically for the emission factor measurements presented in the manuscript. As the authors state, this paper provides critical field measurements of under-characterised combustion aerosol emissions common to developing countries. This is an important subject and within remit of ACP, with its in-the-field online PM emission source measurements. The measurement techniques chosen were appropriate and state-

of-the-science. In general, the authors should be commended on the clear way they have presented their complex results in a manner that should prove useful to future estimates of the air quality impacts of S. Asian combustion sources. I have no major criticisms, but have one or two questions that might be clarified in addition to those of the other reviewer.

line 19 & 31 in the abstract & ...: "Particle phase HCl" - the authors should elaborate on what they think the form of HCl is - hydrochloric acid or organic chloride if its source is chlorinated plastics? and "non-refractory chloride [from BB]" is this condensation of gaseous HCl or a direct primary particulate emission? That the dung-fueled cookstoves were also found to emit ammonium and it was stated that this implied neutralisation, itself implies the form of the chloride as free acid. Since HCl is an extremely volatile gas if not neutralised, is it implied that the particles are simply not in equilibrium?

line 141 "...an attempt to sample at an adequate distance from the point of emissions (typically > 1m) and away from the plume centre to collect cooled and diluted emissions". The authors will appreciate the fact that temperature and dilution control the mass of semi-volatile components in the PM emissions. It is understood that the paper does not aim to provide a detailed analysis of component partitioning, but the authors should provide a brief discussion of their choice of downwind distance (and hence dilution ratio and temperature) and how it will effect the measurement and derived emission factor depending on the volatility profile of the emission and why it is judged to be "more atmospherically relevant" (line 143). This is not to state that these challenging measurements aren't atmospherically relevant, nor that they could have been conducted in a better way, but the possible influences on the reported values should be elaborated on, particularly given the implication of the statement on line 175 that "transmission was ... independent of the dilution factor for non-volatile aerosol". In this regard, the sentence between lines 338 to 341 is interesting in that the poor ventilation in the RETS lab was deemed to invalidate emission factor determination. A brief explanation of the

rationale in this context would probably be warranted.

The discussion of the effects of dilution on the presumably high volatility chloride loading should probably also be included.

Line 322 The contribution to f44 from C2H6N+ is interesting. What is the expected reason for such a high contribution of organic nitrogen from this source? Are there possible implications for f44 from other sources?

Table 1 - expand acronym HW

A clear conclusion of the work might be that the variability across individual source types is significant and raises questions about representativeness of categorisation of emissions within conventional categories used in building gridded inventories. Do the authors have recommendations about how their results should be reflected in inventory categorisation in S. Asia?

---

## Author Comment (AC1) · 31 Aug 2018

The authors thank the referees for their helpful feedback on this manuscript and have listed the comments below with our response or explanation of edits. When possible we have listed the page and line number for any edit.

R1 Comment 1: Because a technical citation for the mini AMS is not given, more of the specific technical details should be listed, in particular the new data acquisition system. Is it the same ADQ system as the new system on newer models of AMS?

Authors: The text has been updated to provide additional technical detail about the

mini-AMS (Page 5 Line 32): "The mAMS has the same vacuum chamber, turbo pump system, and compact time-of-flight mass spectrometer as the Time-Of-Flight Aerosol Chemical Speciation Monitor (TOF-ACSM) (Fröhlich et al., 2013), but contains a chopper system (Jayne et al., 2000) for particle time-of-flight sizing which requires the ADQ data acquisition card. The data acquisition software TOF-AMS DAQ version 5 was used during the campaign. The mAMS used in this work operates with a pseudo-random multi-slit chopper system (ePTOF) that has increased signal to noise ($\sim$50% particle throughput) compared to single slit chopper systems with $\sim$2% throughput, and employs Hadamard Transform for signal inversion (Campuzano Jost, 2014)."

R1 Comment 2: How confident that the Li-Corr factory calibration is still valid? Technical data to back up this assumption should be given, especially because this was operated under very different conditions to the laboratory. Was a post-calibration performed? Were any low pressure calibrations done in the laboratory? Ideally, an uncertainty estimate should be attached to this.

Authors: As stated in the manuscript we had plans to calibrate the gas phase measurements in the field using a calibration standard but were unable to because the standard was contaminated and because the campaign was cut short due to the earthquake. We agree with the reviewer that some assessment of the stability/accuracy of the Li-Corr $CO_2$ measurement is needed. In particular, understanding its relationship with the Picarro $CO_2$ measurement throughout the campaign is critical since the ratio between the two measurements was used to determine the dilution factor. To address this concern, we added an analysis of the relative accuracy between the Picarro and Licor $CO_2$ measurements. The analysis can be found in the supporting information Fig. S2 and shows that under undiluted conditions throughout the campaign the two instruments performed similarly and linear regression results indicate a high accuracy between the instruments (linear slope $\sim$1, $r2 > 0.60$), even under high loading. The absolute accuracy of the Li-Corr or Picarro cannot be assessed, but the consistent relationship between the two indicates that the instruments accuracy did not drift over

time or the unlikely event that the accuracy of the two instruments drifted at the same rate throughout the campaign. Some discussion of these points can be found starting on page 7 line 36: "An intercomparison of undiluted sampling by the Licor monitor and the Picarro $CO_2$ throughout the campaign (Fig. S2) shows that there was no significant drift in relative accuracy between the two instruments over the course of the campaign and that there was no evidence of consistent positive or negative bias during high loadings. Scatter is mostly due to dynamic changes in $CO_2$ concentration which occurs during source sampling. In the more stable overnight monitoring the two instruments showed high correlation ($R^2=0.99$) with a 4% difference in concentration."

R1 Comment 3: The uncertainty estimate attached to the OM:OC estimation based on f44 should be qualified better. Is this precision or accuracy? I would expect the accuracy to be questionable; it is known the relationship between OM:OC and f44 is both instrument and aerosol type specific and there are some types here that are new to the AMS. I would add additional caveats to this effect.

Authors: We agree that the uncertainty attached to the OM:OC results were not clear. The uncertainty in OM:OC was derived from the variability in f44 for each emission source and propagated through the O:C, and OM:OC models and not from error associated with model analysis of f44. A small change to the statement found on page 8 line 33 has been made to make this point more clear: "OA:OC was estimated to have an average uncertainty of $\pm5.3\%$ for the investigated emission sources based on variability in f44 for each emission source (Table 2)."

R1 Comment 4: When describing the Lungdren plots, the authors fail to draw the distinction between continuum/transition aerodynamic used by impactors and vacuum aerodynamic diameters used by the AMS. Which is strange, considering the corresponding author was the first author on the definitive paper on this topic.

Authors: We have addressed this comment by provided conversion factors to aerodynamic diameter and volume equivalent diameter in the supporting information. Additionally on Page 9 Line 31 we have provided the following statement, "While the size cuts used for the Lundgren plots do not perfectly correspond to the aerodynamic diameters measured using impactors due to differences in sizing that are a function of flow regime and density. The approximate conversions to aerodynamic and volume equivalent diameters assuming spherical particles are given in Table S2. However, because we did not have additional aerosol sizing instrumentation in the field we cannot make any interpretations about internal vs external mixing, particle shape, or density. Therefore, conversions from dva to other diameter types such as volume equivalent, or transition regime aerodynamic given in Table S2 are approximate and based on assumptions of sphericity and calculated densities of the aerosol components (DeCarlo et al., 2004). Organic aerosol density was approximated for each emission source using elemental ratios based on work by Kuwata et al. (2011) and can be found in Table 2."

R1 Comment 5: A moderate AAE does not necessarily imply the presence of BrC; BC particles with non-absorbing coatings can also exhibit this, depending on the primary spherule size (Liu et al., Geophys. Res. Lett., 42, 613-619, 10.1002/2014GL062443, 2015). While the authors touch on this, it isn't very clear.

Authors: To address this comment in the methods section we have operationally defined any excess absorption at 370 nm compared to 880 nm as "BrC" but do not make the distinction between light absorbing organics (elsewhere defined as brown carbon) or lensing, to simplify the analysis since we cannot differentiate between the two (Page 7, Line 20). We have added additional discussion about how organic coatings on BC are likely the reason for the observation of "BrC" in the diesel irrigation pump emissions and not light absorbing organics alone and added a reference to Pokhrel et al. (2017) who discussed lensing in detail.

R1 Comment 6: Figure 1:The acronyms next to the mudstoves should be explained in the caption.

Authors: We have edited the caption to read, "Mudstove fuel types are hardwood (Hw), dung (D), and sticks and twigs (Tw)."

R1 Comment 7: Figure 4: Reporting the local time of the measurement is not meaningful. It would be better to use 'time since ignition' as the x-axis.

Authors: We have edited the figure to show time since ignition.

R2 Comment 1: line 19 & 31 in the abstract & ...: "Particle phase HCl" - the authors should elaborate on what they think the form of HCl is - hydrochloric acid or organic chloride if its source is chlorinated plastics? and "non-refractory chloride [from BB]" is this condensation of gaseous HCl or a direct primary particulate emission? That the dung-fueled cookstoves were also found to emit ammonium and it was stated that this implied neutralisation, itself implies the form of the chloride as free acid. Since HCl is an extremely volatile gas if not neutralised, is it implied that the particles are simply not in equilibrium?

Authors: We thank the reviewer for the comment and have made edits to the garbage burning and crop residue burning sections to address with available information what form the HCl emission were in. For garbage burning we suggest that HCl was not in equilibrium, as evidenced by the large gas phase HCl emissions, and that it was condensing into droplets or nucleating as an internal mixture with OA. The similar size distributions of HCl and OA suggest that the HCl was likely internally mixed with OA (Page 11 lines 28-30):

"The small quantity of particle phase chloride under controlled dilution conditions compared to gas-phase HCl under less dilute conditions suggests that condensation of HCl to the particle phase, or the co-condensation of OA internally mixed with HCl, is small in fresh emissions from open garbage burning, but evidence suggests HCl gas migrates to the particles on slightly longer time scales (Liu et al., 2016;Stockwell et al., 2014). Evidence of internal mixing of HCl and OA is observed in the similar size distributions for chloride and OA with the mixed refuse and plastics burning sam-

ples (Figure 2). There remains the possibility that some of the chloride signal is from chlorine-containing organic species, however, this was not observed, and with the non-high resolution mass spectrometer, further detailed work is needed to investigate this possibility."

For crop residue burning, since HCl was not observed in the gas phase and was externally mixed from OA, the HCl was likely neutralized as an inorganic salt. The cation was likely not ammonium though since it was not readily emitted from the crop burning samples (Page 15 lines 30):

"The preponderance of externally mixed, particle-phase chloride suggests condensation of HCl, or nucleation of inorganic salts, is occurring within the crop residue plumes and, unlike what was observed with garbage burning, the inorganic chlorine mass is mostly found in the particle phase. If the particle phase HCl was in the form of inorganic salts instead of a condensed acid, the neutralizing ion was likely potassium and to a lesser extent ammonium (Jayarathne et al., 2018)."

We have already made similar assumptions with and have provided significant discussion in the dung burning section that suggests that the HCl was partially neutralized by ammonium, possibly neutralized by potassium, or in the form of a chloride organic salt (Page 16 line 33):

"Significant chloride and ammonium emissions were also sampled by the off-line filter measurements and gas-phase HCl was not measured above detection limits indicating that particle-phase chloride was dominant with dung burning (Jayarathne et al., 2018;Stockwell et al., 2016). Assuming that all the ammonium measured was a counter ion to the various anion species ($SO_4^{2-}$, $NO_3^-$, $Cl^-$), a predicted ammonium concentration that represents full anionic neutralization was calculated for the NAMaSTE dung-burning samples (RETS samples included). Based on the predicted values, anionic mass from dung burning ranged from 35% to 50% neutralized and the field samples were 45% neutralized. The presence of chloride as the dominant anion in the measured PM1 combined with the lack of HCl observed in the gas-phase, suggests that there was chloride containing organic species present in the dung-burning aerosol, other non-refractory chloride organic salts, or ionic potassium (K+). Slow vaporizing K+ was not observed by the mAMS, but was observed in filter samples to make up an average of 15% of the chloride mass emitted by dung burning (Jayarathne et al., 2018)."

R2 Comment 2: line 141 "...an attempt to sample at an adequate distance from the point of emissions (typically > 1m) and away from the plume centre to collect cooled and diluted emissions". The authors will appreciate the fact that temperature and dilution control the mass of semi-volatile components in the PM emissions. It is understood that the paper does not aim to provide a detailed analysis of component partitioning, but the authors should provide a brief discussion of their choice of downwind distance (and hence dilution ratio and temperature) and how it will effect the measurement and derived emission factor depending on the volatility profile of the emission and why it is judged to be "more atmospherically relevant" (line 143). This is not to state that these challenging measurements aren't atmospherically relevant, nor that they could have been conducted in a better way, but the possible influences on the reported values should be elaborated on, particularly given the implication of the statement on line 175 that "transmission was ... independent of the dilution factor for non-volatile aerosol". In this regard, the sentence between lines 338 to 341 is interesting in that the poor ventilation in the RETS lab was deemed to invalidate emission factor determination. A brief explanation of the rationale in this context would probably be warranted.

Authors: To address this comment we have given some rationale for why the downwind distances and dilution factors were chosen in page 4 line 34 by stating the measurements are atmospherically relevant,"...where the semi-volatile and volatile component are at equilibrium". This definition is obviously subjective but is an attempt to suggest that we tried to measure emissions that have stabilized from the higher concentration and higher temperature conditions of the stack to a more temperature and concentration relevant regime. The outside the stack emissions are closer to an atmospherically relevant thermodynamic and phase partitioning state. We have also included a statement in page 5 line 26 that indicates that we likely measured lower particle phase concentrations of volatile components because of the extent of controlled and uncontrolled dilution compared to if we didn't dilute our samples :

"Transmission of volatile components were not quantified but it is expected that the average dilution factor of 10:1 combined with unquantifiable ambient dilution prior to the inlet, led to lower mass concentrations of semi-volatile organics (Lipsky and Robinson, 2006) and volatile inorganic components (e.g. HCl, H2SO4) in the aerosol phase compared to if the emissions were where not diluted."

Note that this sampling strategy is also discussed by Jayarathne et al. (2018) with similar caveats.

R2 Comment 3: The discussion of the effects of dilution on the presumably high volatility chloride loading should probably also be included.

Authors: Please see the above response to R2 Comment 2 and edits found on page 5 line 26

R2 Comment 4: Line 322 The contribution to f44 from C2H6N+ is interesting. What is the expected reason for such a high contribution of organic nitrogen from this source? Are there possible implications for f44 from other sources?

Authors: We also find the contribution to f44 from amines to be interesting. This will be addressed in more in the Part 2 paper which will discuss the source specific mass spectral results in depth.

R2 Comment 5: Table 1 - expand acronym HW

Authors: We have edited Table 1 and it now reads: "Mudstove fuel types are hardwood (Hw), dung (D), and sticks and twigs (Tw)."

[Figure]

R2 Comment 6: A clear conclusion of the work might be that the variability across individual source types is significant and raises questions about representativeness of categorisation of emissions within conventional categories used in building gridded inventories. Do the authors have recommendations about how their results should be reflected in inventory categorisation in S. Asia?

Authors: This is an excellent question, and one that absolutely requires additional work. The goal of this project was to begin to explore the source emission intensity and composition in South Asia, since it is an area with unique and undercharacterized sources. The choice of categories depends on source-source variation, but also on the inventory application and the available data for source-specific fuel consumption, which sometimes varies seasonally and regionally, etc. We may be able to offer additional guidance in a synthesis paper we hope to produce, but meanwhile, ultimately, this work with the companion papers from Stockwell et al. (2016) and Jayarathne et al. (2018) serve as a start for emission inventories. As noted, there is significant uncertainty that will only be reduced with additional detailed measurements and characterization of emissions. We hope our results both demonstrate the variability, but also serve to provide realistic bounds on emission from poorly characterized sources in an very important and understudied region of the earth.